# Molecular understanding of label-free second harmonic imaging of microtubules

V. Van Steenbergen [1], W. Boesmans [1,2,3], Z. Li[1], Y. de Coene [4], K. Vints[5,6], P. Baatsen[5,6], I. Dewachter[7,8], M. Ameloot [9], K. Clays[10] & P. Vanden Berghe [1,11]

Microtubules are a vital component of the cell's cytoskeleton and their organization is crucial for healthy cell functioning. The use of label-free SH imaging of microtubules remains limited, as sensitive detection is required and the true molecular origin and main determinants required to generate SH from microtubules are not fully understood. Using advanced correlative imaging techniques, we identified the determinants of the microtubule-dependent SH signal. Microtubule polarity, number and organization determine SH signal intensity in biological samples. At the molecular level, we show that the GTP-bound tubulin dimer conformation is fundamental for microtubules to generate detectable SH signals. We show that SH imaging can be used to study the effects of microtubule-targeting drugs and proteins and to detect changes in tubulin conformations during neuronal maturation. Our data provide a means to interpret and use SH imaging to monitor changes in the microtubule network in a label-free manner.

[1] Laboratory for Enteric NeuroScience (LENS), TARGID, Department of Chronic Diseases Metabolism and Ageing, KU Leuven, Leuven, Belgium. [2] Department of Pathology, GROW-School for Oncology and Developmental Biology, Maastricht University Medical Center, 6229 HX Maastricht, The Netherlands. [3] Biomedical Research Institute (BIOMED), Hasselt University, 3500 Hasselt, Belgium. [4] Soft matter and Biophysics, Department of Physics and Astronomy, KU Leuven, 3001 Heverlee, Belgium. [5] Electron Microscopy Platform and VIB Bioimaging core facility, VIB-KU Leuven, 3000 Leuven, Belgium. [6] Center for Brain & Disease Research, Department of Neurosciences, Leuven Brain Institute, KU Leuven, 3000 Leuven, Belgium. [7] Alzheimer Research Group, Biomedical Research Institute, Hasselt University, 3500 Hasselt, Belgium. [8] Institute of Neuroscience, UCLouvain, 1200 Brussels, Belgium. [9] Biophysics, Biomedical Research Institute (BIOMED), Hasselt University, 3500 Hasselt, Belgium. [10] Molecular Imaging and Photonics, Department of Chemistry, KU Leuven, 3001 Heverlee, Belgium. [11] Leuven Brain Institute, KU Leuven, 3000 Leuven, Belgium. Correspondence and requests for materials should be addressed to P.V.B. (email: pieter.vandenberghe@kuleuven.be)

Microtubules are a crucial component of the cell's cytoskeleton. Apart from providing structural support to complex cellular morphologies, they are essential for long distance intracellular transport and mitosis[1–3]. The building blocks of a microtubule are αβ-tubulin dimers, which can polymerize to form fibrillary structures of several micrometers in length. The microtubule network is highly dynamic and remains in a state known as dynamic instability as these biopolymers stochastically switch between growing and shrinking states, never reaching a steady-state length[4]. The microtubule end where α-tubulin is exposed is referred to as the minus end while the growing end, where β-tubulin is exposed, constitutes the plus end[5]. Upon polymerization, the β-subunit is bound to guanosine-5′-triphosphate (GTP) to facilitate incorporation into an existing lattice. Once polymerized, GTP will hydrolyze to form guanosine-5′-diphosphate (GDP)-bound β-tubulin inducing a conformational change in the molecular structure of the tubulin dimer[6–8]. When bound to GTP at the β-subunit, the tubulin dimer conformation is stable and less prone to depolymerization as compared to the GDP-bound conformation. A slight difference in polymerization versus hydrolysis rates ensures a lag-time in favor of the protective GTP-bound tubulin cap at the end of the microtubule, preventing depolymerization[9,10]. Upon loss of the protective cap, microtubules rapidly depolymerize, a process referred to as catastrophe. The microtubule lattice not only contains stable GTP-bound dimers at its growing plus ends, but also at regularly spaced locations along the lattice. These GTP-bound tubulin rich sites are considered rescue sites as they can revert a microtubule that was undergoing catastrophe back into a growing state[11–14]. Apart from being crucial for basic biological functions, perturbations in microtubules and the integrity of the network have been shown to be crucially involved in cancer and the pathogenesis of several neurodegenerative diseases such as Parkinson's and Alzheimer's disease[15,16]. Therefore, understanding how the axonal microtubule network is organized, is not only relevant for basic neurobiology but can also provide insights for translational and clinical research.

Despite the vast interest in microtubule network formation and homeostasis, the technology to study microtubules in living cells remains limited. While fluorescence-based imaging techniques of microtubules are commonly used, they also face an important limitation in that they can either only be applied in fixed cells or interfere with important intracellular processes[17]. Label-free imaging approaches like second harmonic (SH) microscopy can overcome this problem. Upon interaction with an intense short laser pulse, some biomolecules can scatter frequency-doubled photons. The efficiency of this process is dependent on factors like the molecular hyperpolarizability, number, density, and the ordered arrangement of the molecules[18,19]. Because of Neumann's principle, SH generation (SHG) can only occur in molecules and assemblies without a center of symmetry. Although SHG is mainly used to visualize strong biological harmonophores like collagen[19–21], it can also be used to detect the far weaker tubulin dimer as it organizes itself in the microtubule lattice[19,22–26]. Interestingly, though this technique does not require staining or fluorescence-based methods, possibly interfering with cellular processes, the use of SHG microscopy of microtubules is not yet established in neuroscience as the true molecular origin remains elusive and interpretation of the relatively weak signal is not trivial.

In this study, we used SHG to visualize the axonal microtubule network and to facilitate signal interpretation by determining the main factors that contribute to SH signal generation by the microtubule network. Combining SH imaging with correlative fluorescence and electron microscopy (CLEM), we demonstrate the importance of theoretical determinants such as microtubule orientation, number, and organization in biological samples where multiple microtubules are present. Our data show a positive correlation between SH signal intensity and the presence of GTP-bound tubulin dimers. Using Hyper-Rayleigh scattering, we identified the molecular origin of this correlation and found that GTP-bound tubulin has a larger first hyperpolarizability ($β$) compared to GDP-bound tubulin, owing to the conformational changes upon GTP hydrolysis. Finally, we show how SHG imaging of microtubules can be used to study a variety of biological mechanisms in neurodegeneration and neuronal maturation.

## Results

**SHG imaging of microtubules.** SHG is a polarization-dependent coherent nonlinear optical process occurring at exactly the second harmonic wavelength of the incident laser. Therefore, SH signals can be distinguished from other optical phenomena by spectral dispersion and by changing the polarization direction of the laser. To verify that detected signals in neuronal processes are indeed SH, the laser polarization was rotated 90° (Fig. 1a), which only affects SHG and not the fluorescent signal generated by the disorganized cytosolic autofluorescent molecules. Neuronal fibers parallel to the polarization direction show increased signal intensities compared to those perpendicular to the laser polarization. A spectral scan of the nonlinear scattered light was recorded to further prove the SH nature of the signal (Fig. 1b). A narrow peak was detected at exactly half the wavelength of the incoming light while a broad spectrum of longer wavelengths was recorded for the 2-photon generated autofluorescence, which can be used to visualize cell bodies. To validate the microtubular origin of the detected SH signals, we show that also in vitro polymerized microtubules lacking other cellular components are capable of generating SH (Fig. 1c) and that in living cells expressing eGFP-labeled microtubules the fluorescence colocalizes with the SH signals (Fig. 1d). Furthermore colchicine and nocodazole, drugs that destabilize the microtubule network and inhibit microtubule polymerization, respectively, lead to a significant loss of the SH signal (Fig. 1e, f). We further used an axonal and dendritic antibody staining to compare SH signals in both projection types. While we detect SHG in both neuronal processes, correlative SHG-fluorescence analysis shows that the microtubule network in axons is more efficient in generating SHG signals (Fig. 1g), which may explain earlier suggestions that SH would only be generated by axonal microtubules owing to their uniform polarized distribution[24,27]. SHG is a generic property of microtubules and SH microscopy can be applied in a variety of neuronal cell culture systems and neuronal tissues (e.g., brain slices) (Fig. 1h, i).

**Microtubule polarity.** The reason as to why mainly the microtubule network in axons is efficient in generating SH signals has been attributed to the uniform orientation of microtubules in axons[24]. As most microtubules are oriented with their plus ends away from the soma, the resulting SH signals from individual microtubules are parallel-aligned, which causes constructive interference and amplification of the compound SH signal (Fig. 2a). In dendrites, microtubules are not uniformly polarized leading to partial destructive interference (phase mismatching), which may cause the SH signals to drop below detection levels, or in the case of absolute symmetry effectively become zero. To determine the orientation of the microtubules in living neurons, we labeled their plus ends using an EB3-eGFP construct. Live recordings of EB3 and SH imaging was followed by immunohistochemistry to identify the axonal and dendritic processes (Fig. 2b). Based on these live recordings and resulting kymographs, we correlated SH signal intensities with microtubule

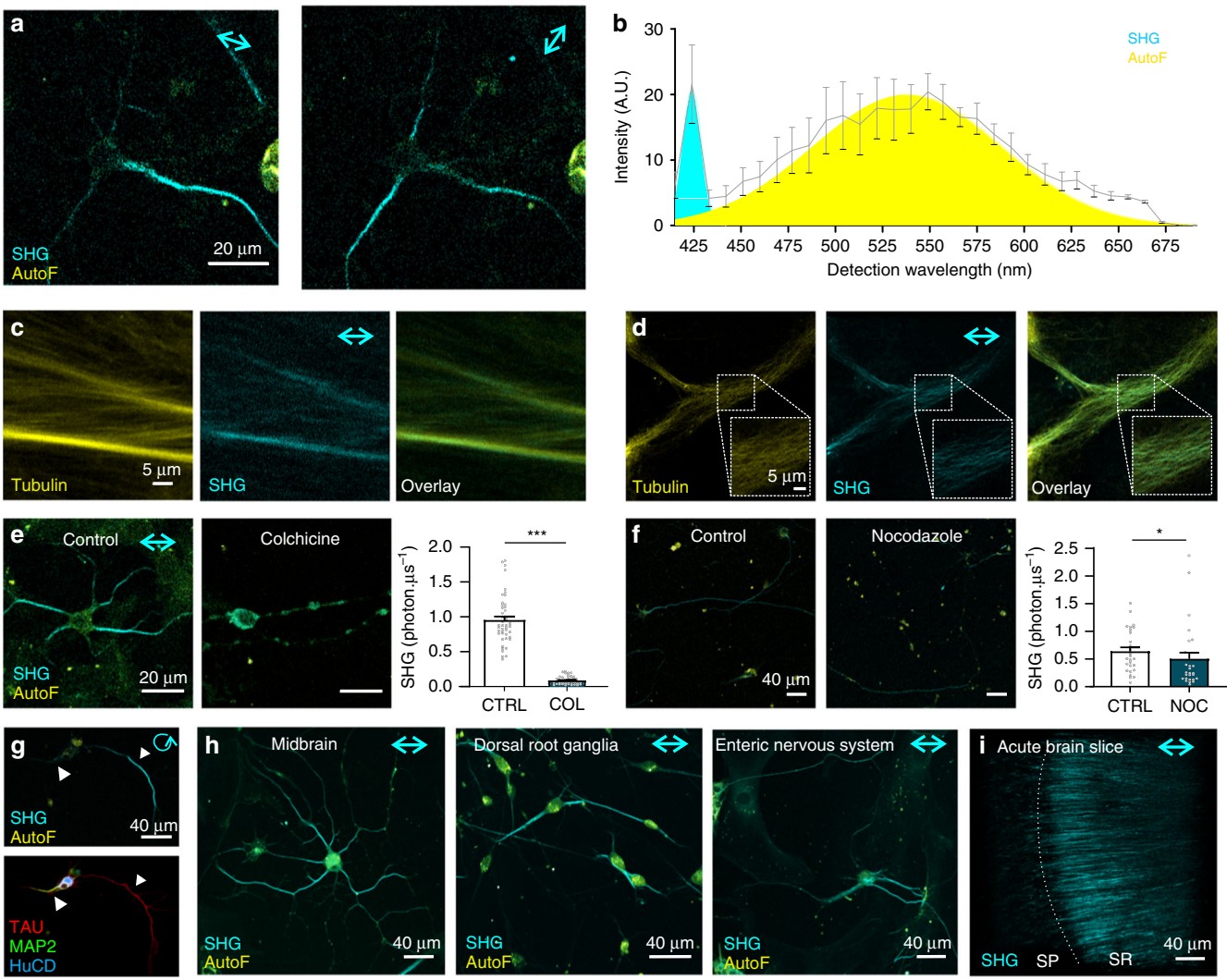

**Fig. 1** SH generation to image microtubules polymerized in vitro, in cell culture and brain slices. **a–f** Confirmation of the microtubular origin of the SH signals. Laser polarization direction is indicated with a double (linear) or circular arrow. **a** Changing laser polarization direction affects the detected SH signal intensity (cyan). The simultaneously recorded autofluorescence signal (AutoF, yellow), is not sensitive to laser polarization. **b** A spectral scan (9 nm wide detection channels) of the emission light in the backward detection path with a narrow peak at the frequency-doubled wavelength of the 850 nm incoming light (cyan peak, 425 nm). Autofluorescence (Yellow Gaussian fit, AutoF) was detected at longer wavelengths ($n = 3$ cells from three independent experiments). **c** SHG signals from in vitro polymerized fluorescently labeled microtubules without any other cellular components, indicating the true microtubular origin of the SH signals. **d** Recordings of neuronal cultures transfected with α-tubulin-eGFP (Yellow) show microtubular localization of the eGFP signal that colocalizes with SH-generating structures (Cyan). **e** SH signals recorded from colchicine (COL, 100 μM, 30 min) treated neurons were significantly decreased compared with untreated (CTRL, dimethyl sulfoxide (DMSO)) neuronal cultures ($n = 27$ cells from three independent experiments; ***$p < 0.001$ Unpaired two-tailed $t$-test). **f** SH signals are significantly reduced in nocodazole (50 nM, 4 h) compared to control (CTRL, DMSO) cells ($n = 27$ cells from three independent experiments; *$p < 0.05$ two-tailed Mann–Whitney test). **g** SH signals in neurons predominantly originate from the axonal microtubule network as indicated by immunofluorescence labeling for TAU (axon, red), MAP2 (dendrites, green), and HuCD (neuronal cell body, blue). **h** Shows three examples of SHG used to visualize microtubules in primary neuronal cell cultures derived from mouse midbrain, dorsal root ganglia and the enteric nervous system. **i** Maximum projection of the SH signals coming from microtubules in axonal processes of coronal acute hippocampal brain slices (250 μm thickness). SP: stratum pyramidale, SR: stratum radiatum. All cultures were imaged at 7 DIV apart from the enteric nervous system cultures at 3 DIV. Bar plots presented as means ± standard error of the mean. All source data are provided as a Source Data file

orientation in both projection types (Fig. 2c). In line with published literature, we find a significant increase in unipolarized microtubules in axons compared to the mixed microtubule polarity in dendrites (Fig. 2d). Apart from microtubule polarity, we also show lower SH signal intensities in dendrites (Fig. 2e). Interestingly, microtubule polarity does not solely determine SH signal intensities as EB3 directionality does not differ between fibers with high (photon count $> 0.35 \mu s^{-1}$) or low (photon count $\leq 0.35 \mu s^{-1}$) SH signal intensities, indicating that also other factors are involved (Fig. 2f).

**Microtubule number and organization.** As a uniformly polarized microtubule network by itself is clearly not yet sufficient to fully explain SHG intensity differences along neuronal projections (see Fig. 2f), we set out to test to what extent two other theoretical determinants for SHG would influence SH signal intensity of microtubules in living neurons. As a low number of microtubules also means fewer SH emitters, a decreased signal is expected compared to fibers with more microtubules, at least when assuming comparable microtubule polarity (Fig. 3a). Indeed, the microtubule SH signal intensity measured in living cells positively

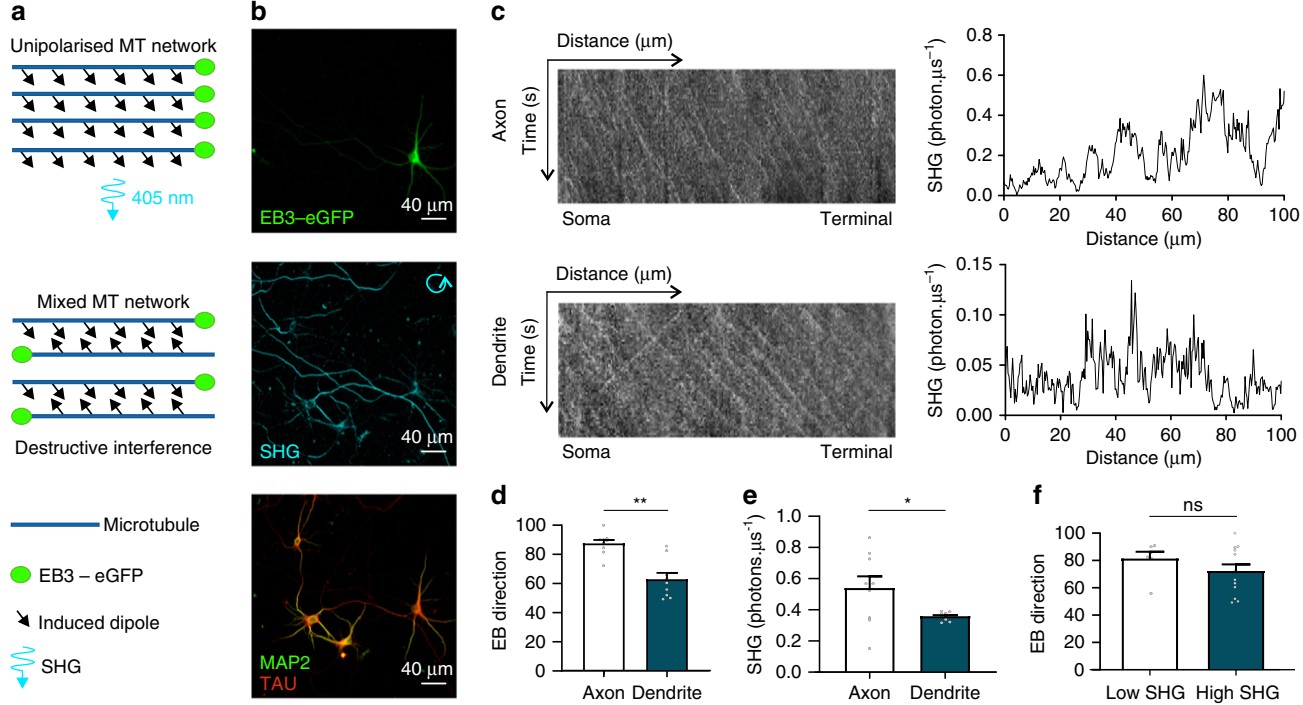

**Fig. 2** Uniform microtubule polarity is not sufficient to generate SH. **a** SH signals from a unipolarised microtubule (MT) network lead to constructive interference and increased SH signal intensities while mixed microtubule polarity results in destructive interference and loss of the signal. EB3 proteins bind to the plus ends of a growing microtubule. **b** First frame of a time-lapse recording of a neuron transfected with EB3-eGFP to determine microtubule orientation, followed by a SHG recording and immunostaining for MAP2 and TAU to differentiate dendritic and axonal projections, respectively. **c** From the kymograph, EB3 directionality and thus microtubule orientation can be determined. Microtubules where EB3 proteins move towards the distal end of the fiber (anterograde movement in the kymograph) have the microtubules with their plus ends out towards the growth cone while EB3 proteins moving towards the cell soma (retrograde movement in the kymograph) have the microtubules oriented with their minus ends out towards the growth cone. Based on the EB3 directionality, the (dominant) microtubular polarity of a fiber was determined followed by a recording of the SH signal intensities along that fiber. A SH intensity plot was generated from the proximal towards the distal end of the neuronal process. **d** Quantification of EB3 directionality, with significantly increased anterograde EB3 comets in axons compared with dendrites ($n = 9$ cells from three independent experiments, **$p < 0.01$, unpaired two-tailed $t$-test). **e** A significant increase in SH signal intensity was measured in axonal projections ($n = 9$ cells from three independent experiments, *$p < 0.05$, unpaired two-tailed $t$-test with Welch' correction). **f** No significant difference was found in the fraction of plus-end out microtubules (anterograde EB3 comets) in fiber regions that generated a high (photon count $> 0.35$ μs$^{-1}$) and low (photon count $\leq 0.35$ μs$^{-1}$) SH signal intensity ($n = 9$ cells from three independent experiments, two-tailed Mann–Whitney test). Neuronal cultures were imaged at 7 DIV. Bar plots presented as means ± standard error of the mean. All source data are provided as a Source Data file

correlated with fluorescence intensities of an α-tubulin immunostaining, used as a measure for the amount of microtubules (Fig. 3b). We used the wide cellular protrusions of some meningeal fibroblasts present in the cultures, making it possible to discriminate between parallel regions generating SH (arrows left panel Fig. 3c). A SH recording from such a fibroblast with three SH-generating regions (as seen in the enlarged image in the top panel of Fig. 3c) was correlated with an electron micrograph of a cross section through the cell to quantify the microtubules present in each SH-generating region (Fig. 3c). The region with increased SH signal intensity had the highest amount of microtubules (94) while the region with low-SH signal intensity contained the lowest microtubule number (36). As detection of SHG heavily depends on constructive interference from multiple microtubules, proper parallel organization of the microtubule network is, in theory, crucial for signal generation (Fig. 3d). Correlative SHG—electron microscopy was used to provide experimental evidence of this theoretical determinant in living cells. Different regions of a neuron generating low (arrow) and increased (arrowhead) SHG signals were correlated with their matched electron micrographs to visualize ultrastructural organization of microtubules (Fig. 3e). In this specific example seven parallel-aligned microtubules are present in the fiber with

increased SH signal intensity and ten, with more random orientation, in the region with a low-SHG signal intensity. The cellular process with low-SHG signals clearly contained a more disorganized microtubule network as compared to the parallel-aligned microtubules found in the intense SH generating process. This is in excellent agreement with the findings of Psilodimitrakopoulos et al.[23] about the importance of parallel organization of the SHG source in neurons.

**Fixation alters SHG and tubulin conformations**. So far, all SH images of microtubules have only been recorded in living cells as commonly used fixatives lead to loss of the signal, as anecdotally mentioned by Dombeck et al.[24] Although disadvantageous from a practical and experimental point of view, scientifically it is very intriguing as to why only unfixed microtubules generate SH, while two other biomolecules, myosin (e.g., in striated muscle) and collagen (e.g., in mouse tail tendon), are not influenced by paraformaldehyde (PFA) fixation in generating SH (Supplementary Fig. 1), indicating that this 'live-only aspect" is specific for microtubules. To understand the true nature of this difference we monitored microtubule SH signals upon addition of fixatives based on drying (acetone, methanol) and chemical crosslinking

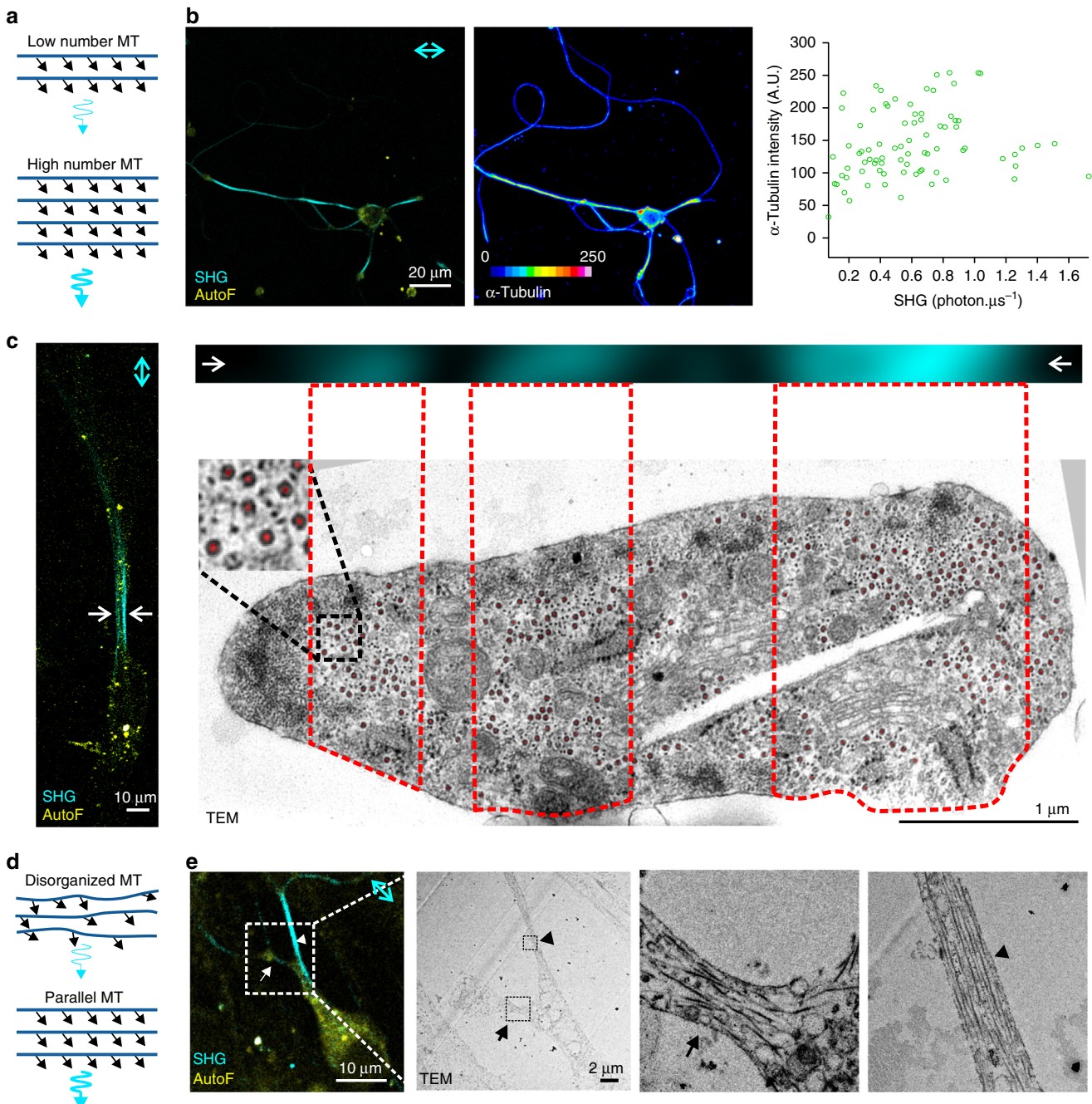

**Fig. 3** Microtubule number and organization influence SH signal intensity. **a** A low number of microtubules will generate SH signals with lower intensities as compared to a high number of microtubules and thus SH emitters. **b** SH signals correlated with α-tubulin staining intensities as a measure for the amount of microtubules present in the fiber ($n = 81$ cells from three independent experiments; $p < 0.01$ Spearman $r = 0.2979$). **c** SHG from three parallel regions in a fibroblast protrusion (arrows in fibroblast overview image (left)) indicate the region enlarged in the image at the top) correlated with a transmission electron micrograph of a cross section to visualize individual microtubules (red dots). Dotted line boxes in the micrograph represent the regions generating SH in the protrusion. **d** In a disorganized microtubule network, the resulting induced dipoles will be less aligned leading to a lower SH signal compared to a network that has more parallel-organized microtubules. **e** A cell with two processes, one that generates low (arrow) and one high (arrowhead) intensity SH signals (left) and its matched correlated transmission electron micrographs (TEM, right) to visualize microtubule organization at the ultrastructural level. The low- and high-intensity SH fibers correspond to a disorganized (arrow) and parallel-organized (arrowhead) microtubule network respectively. Neuronal cultures were imaged at 7 DIV. All source data are provided as a Source Data file

(PFA) and ethylene glycol bis succinimidyl succinate (EGS). We found that all fixatives, apart from EGS, lead to disappearance of the microtubule generated SH signal (Fig. 4a). It is well-known that fixation causes conformational changes in proteins, even to an extent that explains binding affinities of antibodies as their epitope can be altered during the fixation process[28]. A change in molecular conformation may also affect the SH signal as it is linked to the hyperpolarisability and, therefore, the molecular structure of a protein. Since EGS fixation maintains the SH signal, we hypothesized that EGS fixation may preserve a specific tubulin conformation that is abolished by other fixatives. Using an antibody that specifically recognizes GTP-bound tubulin dimers

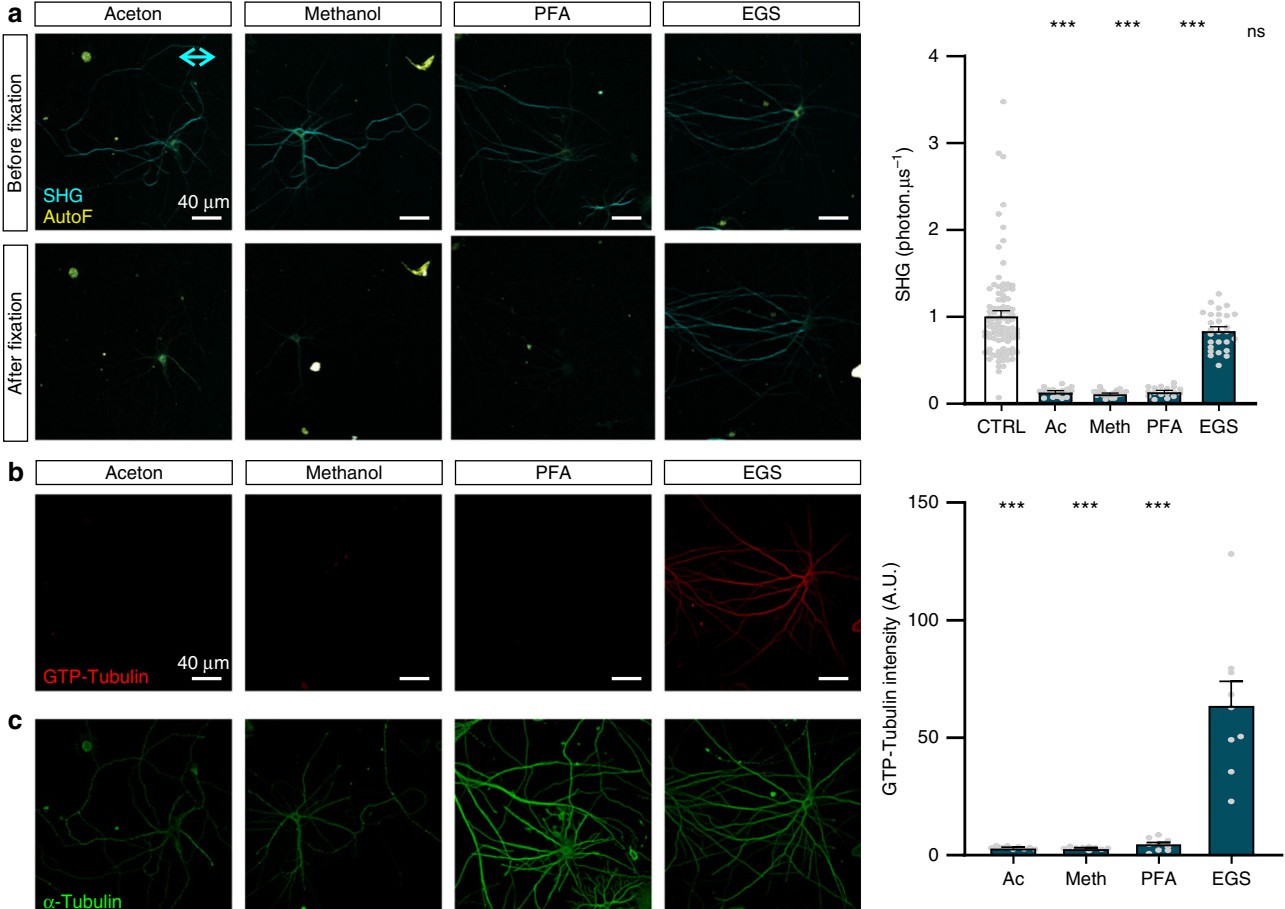

**Fig. 4** Fixation alters SH signals and tubulin conformations. **a** Recordings of SH signals before and after fixation with acetone (Ac, n = 16 cells from three independent experiments), methanol (Meth, n = 20 cells from three independent experiments) and paraformaldehyde (PFA, n = 19 cells from three independent experiments), which all lead to a loss of the SH signal while ethylene glycol bis succinimidyl succinate (EGS, n = 27 cells from three independent experiments) fixation preserves the SH signal intensity compared to the controls before fixation (CTRL; n = 94 cells from six independent experiments; ***p < 0.001 one-way ANOVA Dunn's multiple comparison test). **b** MB11 staining and quantification for GTP-bound tubulin dimers after fixation shows lack of epitope recognition in aceton, methanol, and paraformaldehyde fixed cells, but positive staining in EGS fixed samples (n = 9 cells from three independent experiments; ***p < 0.001 one-way ANOVA Bonferroni multiple comparison). **c** To control for immunohistochemical processing, staining with α-tubulin was performed simultaneously and shows positive staining after aceton, methanol, PFA, and EGS fixation. Neuronal cultures were imaged at 7 DIV. Bar plots presented as means ± standard error of the mean. All source data are provided as a Source Data file

(MB11), we show that the GTP-bound tubulin dimer conformation is only present in EGS fixed cells but not in acetone, methanol or PFA fixed samples (Fig. 4b). An α-tubulin antibody was used to quantify microtubule content and to confirm the efficiency of immunofluorescence labeling procedures in each of the fixation methods (Fig. 4c). While GTP-bound tubulin specific stainings were only present in EGS fixed cells, α-tubulin labeling was positive for each of the fixatives and even more prominent in PFA fixed neurons (compared to EGS), explicitly confirming the conformational specificity of the GTP-bound tubulin antibody.

**GTP-bound tubulin dimer conformation.** As EGS fixation preserves the SH signal and the GTP-bound tubulin dimer conformation, we tested the hypothesis that the SH signal from microtubules mainly originates from this specific molecular structure. In a first set of experiments, we found a positive correlation between SH signal intensity and MB11 staining intensity as a measure for the amount of GTP-bound tubulin dimers (Fig. 5a). The resulting Spearman correlation between SH and the tubulin staining intensity increased when a GTP-specific rather

than a general α-tubulin antibody was used (Fig. 3a). Furthermore, a power fit through both data sets shows that the GTP-bound tubulin data obey more by the theoretical 0.5 power relation with SH than the α-tubulin data (Supplementary Fig. 2). We show that apart from neurons, the SH signals from microtubules in fibroblasts mainly originate from MB11-positive microtubules in constricted areas where the GTP-bound tubulin islands are abundant and parallel aligned (Supplementary Fig. 3). Taxol, which stabilizes the microtubule network by increasing the GTP-bound tubulin dimer conformation[11,29] (Supplementary Fig. 4), increased the SH signal intensity in neuronal cultures compared to controls (Fig. 5b). These results are comparable when using epothilone B, having a similar mode of action as taxol (Fig. 5c). HEK239T cells are not efficient in generating SH signals (Supplementary Fig. 5A) as their microtubules are not uniformly polarized, have a lower packing density compared to axons and are less parallel. However, changing their tubulin conformation by addition of taxol, made even those microtubular networks visible with SHG imaging (Supplementary Fig. 5A). A confounding aspect is the fact that taxol itself forms crystalline aster-like SH-generating structures. To confirm the microtubule origin

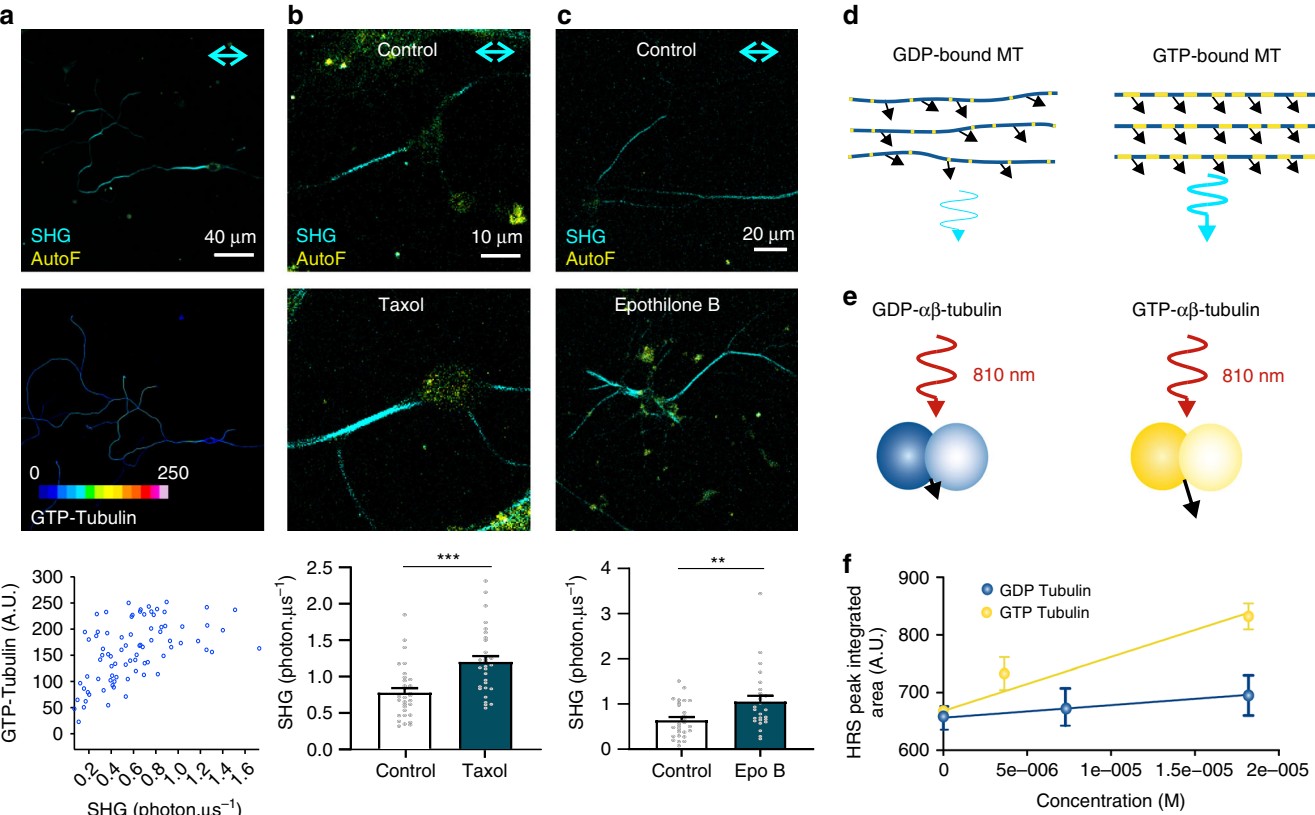

**Fig. 5** SH signals originate from the GTP-bound tubulin dimer conformation. **a** SH signals positively correlate with MB11 (GTP-bound tubulin conformation) staining intensity ($n = 81$ cells from three independent experiments; $p < 0.001$ Spearman $r = 0.5642$). **b** Increased SH signal intensities in untreated (DMSO) and taxol (10 nM, 4 h incubation)-treated hippocampal neurons ($n = 27$ cells from three independent experiments ***$p < 0.001$ two-tailed Mann–Whitney test) and HEK293T ($n = 27$ cells from three independent experiments; ***$p < 0.001$ Welch two-tailed $t$-test) cells. **c** SH signals were significantly increased upon addition of epothilone B (10 nM, 6 h) in neuronal cultures ($n = 27$ cells from three independent experiments; **$p < 0.01$ two-tailed Mann–Whitney test. **d** Schematic representation of how increased tubulin dimers in the GTP-bound conformation (yellow compared to blue GDP-bound tubulin conformation) could impact microtubule organization and SH signal intensity. The GTP-bound conformation leads to more rigid, less bendable microtubules, which in turn lead to a more parallel organization. **e** Because of the molecular alterations between GDP- and GTP-bound tubulin dimers, the dipolar hyperpolarisability tensor element $\beta_{zzz}$ (arrow) can be different, which directly affects SH signal intensities. Note that the arrows here represent the hyperpolarizability tensor element. **f** HRS shows a larger $\beta_{zzz}$ for GTP-bound tubulin dimers ($\beta_{zzz} = 190 \pm 30 \times 10^{-30}$ esu) compared to GDP-bound tubulin dimers ($\beta_{zzz} = 100 \pm 10 \times 10^{-30}$ esu), as calculated from the slope and intercept of the concentration dependency of the HRS peak amplitude. $n = 10$ measurements per concentration (mean ± standard deviation). Neuronal cultures were imaged at 7 DIV. Bar plots presented as means ± standard error of the mean. All source data are provided as a Source Data file

of the taxol-induced SH signal increase, we added PFA fixative, which abolished the cellular SH signals but not those generated by the taxol asters (Supplementary Fig. 5B). Since GTP-bound tubulin dimers lead to more rigid microtubules, less prone to bending[30,31], two explanations can be given for the positive correlation between GTP-bound tubulin dimers and SHG. Either, the increased rigidity of the microtubules affects their organization, increasing the number of parallel-aligned dipoles and thus enhance SH signal intensities (Fig. 5d). Another possibility is that, since the molecular structure of GTP and GDP-bound tubulin dimers differs, the dipolar first hyperpolarisability tensor element $\beta_{zzz}$ of GTP-bound tubulin is larger compared to GDP-bound tubulin dimers (Fig. 5e). Therefore, we used hyper-Rayleigh scattering (HRS) to determine this hyperpolarisability tensor element of both dimer conformations (Fig. 5f and Supplementary Fig. 6). GTP-bound tubulin dimers indeed have an increased hyperpolarisability ($\beta_{zzz} = 190 \pm 30 \times 10^{-30}$ esu) compared with GDP-bound tubulin dimers ($\beta_{zzz} = 100 \pm 10 \times 10^{-30}$ esu, $n = 10$ measurements per concentration). Both GTP and GDP exhibit HRS depolarization ratios close to 5 ($4.9 \pm 0.3$ and $4.4 \pm 0.3$, resp.), which indicates that the dipolar tensor component is

dominant indeed, yet that the GTP has a slightly better dipolar ordered structure[32]. Note that a factor of 2 larger hyperpolarizability for GTP translates in a factor of 4 larger second harmonic intensity. The elevated hyperpolarizability of the GTP-bound tubulin dimer, makes this conformation an important and strong contributor to the overall SH signal from microtubules.

**Biomedical applications for SH (imaging) microscopy.** SHG microscopy is a powerful technique applicable in living cells that allows imaging microtubule composition in a label-free way. We show that with SHG one can extract information from the microtubule network at the molecular level while cells are alive. Despite the attractive characteristics, SHG of microtubules is not widely used in biology or biomedicine, as the determinants are not well understood. Now, we show three distinct applications of how SH imaging can be used in a biomedical setting as a sensitive imaging technique for microtubules to analyze their number, organization, and molecular conformation. In a first set of experiments, we show that the SH signal from *substantia nigra* neurons is lost upon incubation with rotenone, a drug that binds

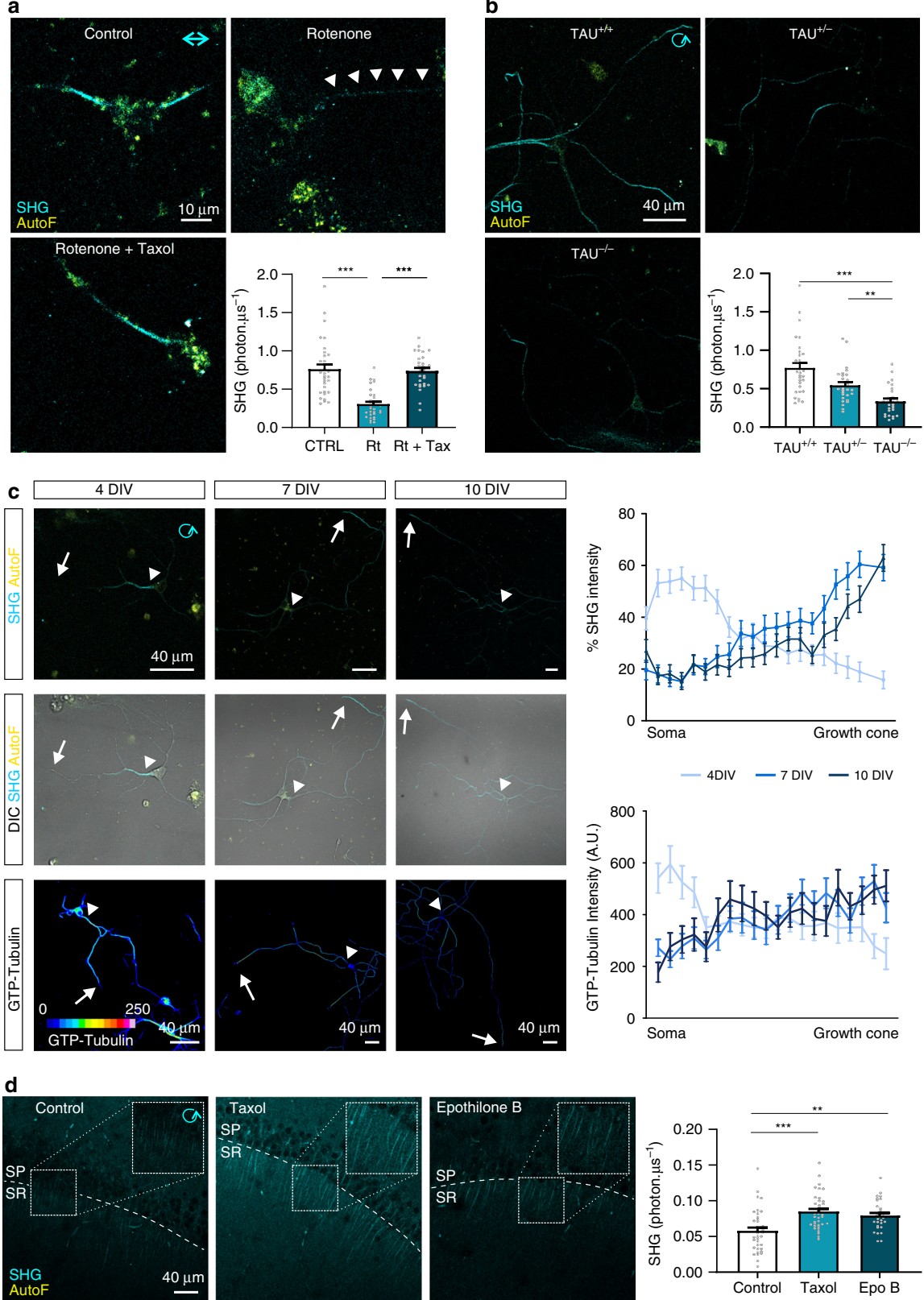

the same site as colchicine and destabilizes microtubules (Fig. 6a)[33]. As co-treatment with taxol has been shown to rescue microtubule stability[34], we set out to study whether this rescue effect is also detectable in living cells using SHG rather than immunohistochemistry in fixed cells. Rotenone drastically

reduced SH signal intensity in neuronal fibers, which was fully prevented in neurons that were co-treated with rotenone and taxol (Fig. 6a). Furthermore, we show that the SH signal generated by microtubules in neurons deficient for Tau (TAU$^{-/-}$), a microtubule associated protein that bundles microtubules[35], is

**Fig. 6** Biomedical applications for SH imaging of microtubules. **a** SH recordings of primary neuronal cultures from the *substantia nigra* treated with DMSO as control (CTRL), rotenone (Rt 100 nM, 4 h) or a combination of rotenone (100 nM, 4 h) and taxol (100 nM, 4 h). The significant reduction in the SH signal upon incubation with rotenone (arrowheads) was no longer detected in neurons treated with both rotenone and taxol to counteract the destabilizing effect ($n = 27$ biologically independent cells; ***$p < 0.001$ one-way ANOVA Dunn's multiple comparison test). Neuronal cultures were imaged at 7 DIV. **b** SH signals from hippocampal neuronal cultures prepared from wild-type (TAU$^{+/+}$), heterozygous (TAU$^{+/-}$) and TAU knock-out (TAU$^{-/-}$) mice. The SH signal intensity is significantly lower in TAU-deficient neurons compared to control and heterozygous cells ($n = 27$ cells from three independent experiments; **$p < 0.01$ ***$p < 0.001$ one-way ANOVA Dunn's multiple comparison test). Neuronal cultures were imaged at 7 DIV. **c** Hippocampal neurons were cultured for 4, 7, or 10 days in vitro (DIV) before SH recordings and MB11 staining to localize GTP-tubulin dimer conformations. The SH signal and MB11 staining intensity was plotted over the entire length of the fiber to detect regional differences along the fiber during neuronal maturation in vitro. At early stages of development (4 DIV), both the SH and MB11 signal intensity were mainly localized at the proximal region of the processes (arrowhead) while later stages (7, 10 DIV) show increased signal intensity at the distal end (arrow) of the fiber ($n = 27$ cells from three independent experiments; ***$p < 0.001$ two-way ANOVA Bonferroni post-hoc test). **d** SHG recordings in acute brain slices treated with taxol (10 μM, 4 h; $n = 36$ fibers from three independent experiments), epothilone B (100 nM, 4 h; $n = 31$ fibers from three independent experiments) or DMSO ($n = 36$ fibers from three independent experiments) as control show increased SH signal intensities in both taxol (***$p < 0.001$, one-way ANOVA Bonferroni multiple comparison) and epothilone B (**$p < 0.01$, one-way ANOVA Bonferroni multiple comparison) treated slices. SP: stratum pyramidale, SR: stratum radiatum. Bar plots presented as means ± standard error of the mean. All source data are provided as a Source Data file

significantly reduced compared to control (TAU$^{+/+}$) neurons (Fig. 6b), even though the total amount of microtubules was not found to be different in the post-hoc immunohistochemical staining for tubulin (Supplementary Fig. 7). The SH measurements prove to be highly sensitive to Tau induced organization as an intermediate SH signal intensity is measured in the heterozygous (TAU$^{+/-}$) neurons. We also confirmed that the presence of green-fluorescent protein in the Tau-deficient cells does not interfere with SH signal intensity as there is no shift in the ratio of SH signals between 810 or 850 nm excitation, changing the excitation more towards the GFP absorption spectrum (Supplementary Fig. 8). Finally, we followed SH signal localization along the axon and compared neurons during maturation at 4, 7, and 10 days in vitro (DIV) (Fig. 6c). During early stages of neuronal maturation in vitro, the SH signal intensity was highest at the proximal part of the process while at later stages the highest intensity shifts towards the distal end and growth cone of the fiber. Using MB11 stainings we found that GTP-bound tubulin dimers were also present at the proximal axon at 4 DIV while for neurons further along in their development GTP-bound tubulin dimer levels are restricted to the distal end. SH imaging of microtubules is not only possible in primary neuronal cultures but also efficient in brain slices (Fig. 1i). Acute hippocampal slices were incubated in taxol, epothilone B, or dimethyl sulfoxide (DMSO) as a control before recording SH signals in the CA1 stratum radiatum, where projections run parallel (Fig. 6d). We measured a significant increase in SH signals from microtubules in both taxol and epothilone B treated slices compared to controls.

## Discussion
The microtubule network is a vital component of the cytoskeleton and it is critical for most of the trafficking that occurs inside a cell and along its processes. Understanding how the microtubule network is organized and fine tuned is not only important to fully comprehend neural function but is also relevant for elucidating the mechanisms underlying neurodegenerative diseases and axonal regeneration[15,16,36]. Visualization of the microtubule network in living cells, mostly relies on exogenous labels such as fluorescent proteins and dyes, which undoubtedly interfere with intracellular processes. Therefore, we propose the use of SH imaging microscopy as a label-free tool to visualize microtubules[18], microtubular dynamics, and drug-induced alterations. We show that apart from theoretical SH determinants such as number and order, the SH signal mainly originates from one specific and stable GTP-bound tubulin dimer conformation

located at microtubule plus ends and along the microtubule lattice where they serve as rescue sites[11,12,37].

The current understanding of how neuronal fibers generate detectable SH signals is limited, and boils down to the concept that the SH phenomenon is only present in case of uniform polarity of microtubules in axons, mature dendrites, and mitotic spindles in non-neuronal cells[24–27,38–40]. However, we show that, in case sufficient detection sensitivity is available, SHG imaging of non-neuronal cells such as fibroblasts is not only limited to detecting microtubules in the spindle apparatus during cell division. A second argument against microtubule polarity being the sole determinant, resides in the fact that the intensity of the SH signal is not uniformly distributed along axonal processes, while overall microtubule polarity is thought to be consistent[41–44]. Using an end-binding assay, we investigated the relationship between SH signal generation and polarity of the microtubules. While uniform polarity is a determinant to generate detectable SH signals, it is not sufficient to explain the intensity differences in neuronal projections.

Two other determinants that can be deducted from SH theory are the number and organization of microtubules, especially as SHG is a coherent process and detection of a signal strongly depends on efficient constructive interference. This leads to the assumption that the more microtubules are present, the more SH emitters can be detected from and therefore the higher the SH signal intensity. We indeed found a positive correlation between the SH and tubulin staining intensity, as a measure for the amount of microtubules present in the fiber. We took advantage of the wide protrusions of fibroblasts to investigate how SH signal intensity relates to microtubule number in the same cell. Using correlative SH-electron microscopy we show that regions with high SH signals indeed correlate with large microtubule numbers compared to regions with lower SH signals. Interestingly, we also found fibers with a high-tubulin staining intensity but no detectable SH signal, again indicating that the SH signal generation is not solely dependent on microtubule number. As constructive interference relies on proper alignment of the resulting induced dipoles on the molecular level, and thus on parallel-organized microtubules, we compared SH signal intensity with the organization of the microtubule network using correlative SH-electron microscopy. A fiber generating high-intense SH signals had a parallel-organized microtubule network while another fiber from the same cell generating far lower SH signals indeed showed a disorganized microtubule network.

A known but so far unresolved effect of paraformaldehyde fixation is that the SH signal is lost. We show that this is not only

the case for fixation based on chemical crosslinking, but also for acetone and methanol fixation procedures. Interestingly, PFA fixation does not affect the SH signal coming from other biomolecules such as myosin and collagen, indicating that the fixation effect is specific for microtubules and tubulin proteins. This cannot be explained by any of the three theoretical determinants shown earlier as neither microtubule polarity, number or organization can be affected during fixation to such an extent as to cause the loss of the signal. However, fixation methods can alter, in a compound specific way, the conformation of proteins and thus epitope recognition by antibodies[45–47]. When EGS was used as fixative, the SH signal can be preserved, which indicates the involvement of a subtle though crucial molecular state. Two distinct types of tubulin conformations exist in living cells, either stable when bound to GTP or unstable when bound to GDP[1,48]. Dimitrov et al.[11] have developed an antibody that specifically recognizes GTP-bound tubulin dimers but found that it only binds in living cells. Combining our findings on the fixation effects of SH and their work on the loss of the GTP-bound tubulin dimer conformation in fixed cells, we proceeded with testing the GTP-tubulin antibody in our fixed samples. We indeed found that PFA fixation, identical to acetone and methanol fixation, hinders epitope recognition by the antibody. EGS fixation however, does not interfere with binding of the GTP-bound tubulin antibody, indicating that this specific conformation of tubulin is still present in EGS fixed cells, possibly because of the longer spacer arm length.

As EGS fixation preserves the SH signal and the GTP-bound tubulin dimer conformation, we hypothesized that our SH signal could mainly originate from the stable tubulin dimer conformation. First, we found a positive correlation between the SH signal intensities and the GTP-tubulin staining intensities in neuronal processes. The correlation coefficient between SHG signal intensities and GTP-bound tubulin dimers is higher than for the overall α-tubulin signal. Further, we show that SH signals occasionally detected in fibroblasts also originate from microtubules that stained for GTP-bound tubulin. To assess the influence of GTP-bound tubulin on the SH signals, we modified the GTP-bound conformation using taxol. Taxol stabilizes the microtubule network by locking the tubulin dimer in the stable GTP-bound conformation[29]. Neurons incubated with taxol show increased SH signal intensities. Apart from primary nerve cell cultures, also HEK293T cells show significant increases in SH signal intensity upon taxol treatment. As HEK293T cells are not efficient in generating SH from their microtubules since they are not uniformly polarized, not densely packed nor organized in a parallel manner, the increase in the SH signal after changing the tubulin dimer conformation substantiates the importance of tubulin conformation changes for microtubule SHG. The reason for the correlation between SH signals and GTP-bound tubulin dimers could be twofold. On the one hand, this stable tubulin dimer conformation leads to very rigid microtubules, inducing a straight rather than bent organization[30,31]. As this change in organization leads to parallel microtubules, increased signals would be detected owing to more efficient constructive interference. On the other hand, because SH signals are based on the hyperpolarisability of the protein, the change in tubulin conformation from GTP to GDP-bound tubulin could also alter its tensor, leading to changes in the detected SH signal. To test the latter, we used HRS to determine the dipolar hyperpolarisability element of both GTP and GDP tubulin dimers. We found that GTP-tubulin dimers indeed had a larger $\beta_{zzz}$ compared to GDP tubulin. This relates to the molecular changes in the dimer as GTP-bound tubulin dimers have less antiparallel beta-sheets and a more ordered helix as compared to GDP-bound tubulin[49]. This finding shows

that apart from the organizational differences between microtubules containing GTP or GDP-bound dimers, the SH signal can be used to detect changes in tubulin dimer conformations at the molecular level. A study by Psilodimitrakopoulos et al. looking into the effect of oxygen and glucose deprivation in axons shows a change in the SHG signal response to changes in the laser polarization[50]. Using a three-dimensional polarized SHG biophysical model, they propose different structural changes including the conversion of GTP- towards GDP-bound tubulin dimers as a possible cause for the change in SHG response.

Finally, we show how the use of SHG imaging microscopy can complement commonly used imaging techniques. As microtubules have to be either fixed (not allowing to obtain temporal information from microtubule dynamics) or fluorescently labeled (adding a significant weight and limiting the recording length due to toxicity), a technique that allows label-free imaging of microtubules in living cells would be of great benefit. We demonstrate that by using drugs that interfere with microtubules, either destabilizing (rotenone) or stabilizing (taxol), SH imaging is sensitive enough to be used and detect changes in the dynamic microtubule network. Furthermore, because fluorescent labeling is not required, these cells could be followed over longer periods of time to gain temporal information on the effect of a microtubule-acting drug. As Tau is involved in neurodegenerative diseases such as Alzheimer disease, as the most prevalent member of a complete family of tauopathies, the generation of Tau knockout mice was a promising approach to further elucidate the role of Tau in the organization of the microtubule network[51]. Interestingly, no change in phenotype or behavior was observed, nor were any changes in the microtubule network, at least based on fluorescence microscopy, reported[52]. Only electron microscopy in fixed samples revealed small differences[53]. We demonstrate the high sensitivity of SH as an imaging method as we detect a significant difference in the SH signal intensity, but not in the overall tubulin levels. Since no change in tubulin number or conformation was found in Tau-deficient neurons, we propose that the change in the signal is linked to subtle changes in the organization of the microtubule network as suggested by previous electron microscopy studies. Contrary to losing SH by removing Tau, addition of Tau amplifies the SH signal from microtubules as was shown by Stoothoff et al.[22] in an experiment using overexpression of wild-type TAU. Next, we followed the SH signal during neuronal maturation in vitro and traced its intensity along the fiber. SH signals were most intense at the beginning of the neuronal process at 4 DIV while the signal shifted towards distal parts of the fiber at later stages (7 and 10 DIV). Apart from the SH signal, also the presence of the stable GTP-bound tubulin dimer coincides with the localization of the SH signal. As GTP-bound tubulin dimers represent a stable conformation, not prone to depolymerization, we propose that the increased presence of GTP-bound tubulin and thus increased SH signals at the proximal end of the fiber at early developmental stages reflects a stabilization phase of the fiber before further maturation. Once differentiated into axonal and dendritic processes, stabilization would no longer be required at the established proximal part while the still extending distal process contains more growing microtubule ends and thus GTP-bound tubulin dimers. Finally, to illustrate the broad applicability of SH imaging in the neuroscience field, we imaged microtubules in living brain slices of the hippocampus. As a proof of principle we incubated slices with microtubule stabilizers taxol and epothilone B and measured a significant increase in SH signal intensity, similar to neuronal cultures. The practical use of SH imaging is therefore not limited to microtubules in neuronal cultures but can also be applied in acute or organotypic brain slice models.

In summary, in contrast with current models, we found that merely the uniform polarization of the microtubule network in axons is not sufficient to generate a detectable SH signal. Apart from microtubule polarity, also the amount of microtubules and especially their organization influence the signal. Using HRS, we show that the SH signal generated by microtubules mainly originates from stable GTP-bound tubulin dimers, owing to its larger dipolar hyperpolarisability. We show that SH imaging can be used to investigate the effect of microtubule-interacting drugs and that SH imaging is a more sensitive method compared to fluorescence microscopy for detecting subtle changes in the organization of the microtubule network. We demonstrate the ability of SH imaging to detect conformational changes in the tubulin dimer whose localization in a process seems to be tightly regulated during neuronal growth. Finally, our unique insights in label-free detection of microtubules open up avenues for the study of crucial biological processes such as neuronal development and most importantly also provide a tool to monitor the efficacy of microtubule-interacting drugs and study the pathogenic process of neurodegenerative disorders.

## Methods

**Primary neuronal cultures.** All procedures were approved by the Animal Ethics Committee of the University of Leuven. Hippocampal cultures were used for all experiments unless stated otherwise.

*Hippocampal cultures*: Postnatal day 0–5 C57Bl/6J and Mapt$^{tm1(EGFP)Klt}$/J[54] mice pups were quickly decapitated before dissection. Hippocampi were dissected in sylgard dishes containing cold sterile Hank's Buffered Salt Solution (HBSS in mM: 5.33 KCl, 0.44 KH$_2$PO$_4$, 137.93 NaCl, 0.34Na$_2$HPO$_4$.7H$_2$O, 5.56 D-glucose and 10 HEPES. The tissue was minced into small pieces and incubated in 0.25% trypsin-EDTA (Gibco) supplemented with 80 U ml$^{-1}$ DNAse (Roche) for 10 min at 37 °C. After three consecutive wash steps with HBSS supplemented with 10% fetal bovine serum (FBS, Sigma-Aldrich), the tissue was mechanically dissociated by trituration with syringes with decreasing diameter. Cells were plated at $5 \times 10^5$ cells per coverslip (18 mm diameter, coated with poly-D-Lysine) and grown in a 37 °C, 5% CO$_2$ incubator in Neurobasal-A media (Thermo Fisher Scientific) supplemented with 0.5% penicillin/streptomycin (Lonza), 0.5% B27 (Gibco), 100 ng ml$^{-1}$ nerve growth factor (Alomone Labs) and 2 mM Glutamax (Thermo Fisher Scientific). Media was replaced 1:1 every 3 days and cells were used at 7 DIV unless stated otherwise.

*Enteric nervous system (ENS) cultures*: The preparation of ENS cultures was adapted from Boesmans et al.[55]. After cervical dislocation, the ileum of 2-month-old C57Bl/6J mice was isolated and placed in ice-cold sterile oxygenated (95%O$_2$/5%CO$_2$) Krebs solution (in mM: 120.9 NaCl, 5.9 KCl, 1.2 MgCl$_2$, 1.2 NaH$_2$PO$_4$, 14.4 NaHCO$_3$, 11.5 glucose, 2.5 CaCl$_2$). Longitudinal muscle strips with myenteric plexus were washed extensively and cut into small pieces in sterile Krebs solution before digestion in 1.3 mg ml$^{-1}$ collagenase type II (Sigma-Aldrich), 2 mg ml$^{-1}$ protease (Sigma-Aldrich) and 0.3 mg ml$^{-1}$ bovine serum albumin (Life Technologies) for 60 min at 37 °C. After a short centrifugation step (8 min, $360 \times g$) the pellet was resuspended in 0.05% Trypsin-EDTA (Gibco) in HBSS (7 min, 37 °C). Enzymatic digestion was stopped in DMEM/F12 medium containing 10% FBS and the tissue was centrifuged (8 min, $360 \times g$). Cells were resuspended in complete medium consisting of Neurobasal-A medium supplemented with 0.5% penicillin/streptomycin, 2% B27, 1% FBS, 2 mM L-Glutamine (Gibco) and 10 ng ml$^{-1}$ glial cell line-derived neurotrophic factor (GDNF, Invitrogen). A 40 μm mesh cell strainer was used to remove debris before plating with complete medium on precoated coverslips (18 mm diameter, coated with poly-D-lysine).

*Dorsal root ganglion cultures*: DRG cultures were prepared from 2-month-old C57Bl/6J mice. After cervical dislocation, DRGs were isolated and placed in ice-cold basal medium consisting of Neurobasal-A medium supplemented with 10% FBS. Enzymatic digestion with Collagenase type I (0.2 mg ml$^{-1}$, Thermo Fisher Scientific) and Dispase (0.25 mg ml$^{-1}$, Sigma-Aldrich) occurred at 37 °C for 40 min on a shaker (40 rpm). After extensive washing in basal medium, cells were mechanically dissociated using needles of decreasing diameter. Cells were plated (18 mm diameter, coated with poly-D-Lysine and laminin) in Neurobasal-A medium supplemented with 0.5% penicillin/streptomycin, 1% B27, 2 mM Glutamax and 2 ng ml$^{-1}$ GDNF.

*Dopaminergic enriched cultures*: Postnatal day 0–5 C57Bl/6J were quickly decapitated before dissection. The midbrain region containing the ventral tegmental area (VTA) and substantia nigra (SN) were dissected in sterile dissection buffer (in mM: 136.9 NaCl, 5.4 KCl, 0.8 NaH$_2$PO$_4$, 0.2 KH$_2$PO$_4$, 34.7 glucose, 8.8 sucrose, 12.3 HEPES). The tissue was minced into small pieces and incubated in papain (Roche) for 5 min at 37 °C and 5 min at room temperature (RT). Three wash steps were performed using dissection buffer supplemented with DNase I (80 U ml$^{-1}$, Roche). A final wash step was executed using plating media (DMEM-F12 (ATCC), 0.5% penicillin/streptomycin, 2 mM L-Glutamine, 5% horse serum,

and 0.5% B27. Cells were plated with B27 plating media at $5 \times 10^5$ cells per coverslip (18 mm Ø, coated with poly-D-lysine) and maintained at 37 °C in 5% CO$_2$. Media was replaced after the first day and subsequently every 2 days.

**Acute brain slices.** Acute brain slices were prepared from a 2-month-old Wnt1-Cre; R26R-GCaMP6f mouse, expressing the genetically encoded Ca$^{2+}$ indicator GCaMP6f in neural crest-derived cells and defined regions in the brain such as the hippocampus[56,57]. The brain was removed after cervical dislocation and embedded in 4% low-melting point agarose. Ice-cold artificial cerebrospinal fluid (ACSF) (in mM:125 NaCl, 2.5 KCl, 1 MgCl$_2$, 1.25 NaH$_2$PO$_4$, 2 CaCl$_2$, 25 NaHCO$_3$, 25 dextrose), oxygenated by bubbling with 95%O$_2$/5%CO$_2$, was used to fill the vibratome well for slicing. Coronal 300 μm slices were cut on an Integraslice 7550 MM vibratome (Campden Instruments, kindly made available by Thomas Voets, KULeuven) before transfer to a custom-made microscopy holder for SH recordings.

**Transfection.** After 6 DIV, expression of eGFP-tagged α-tubulin (gift from Mathieu Bollen) or eGFP-tagged end-binding protein (gift from Sebastian Munck) was induced in neuronal cultures by using Lipofectamine 2000 (Invitrogen). Per well, 1 μg tubulin plasmid or 0.5 μg EB3-eGFP plasmid was mixed with 0.02% Lipofectamine reagent in Neurobasal-A media and incubated at room temperature for 30 min. The mixture was added dropwise to the wells and expression was checked the following day.

**Pharmacological treatment.** Colchicine (100 μM, 30 min, Sigma-Aldrich), Nocodazole (50 nM, 4 h, Sigma-Aldrich), and DMSO (controls) were applied in plating medium to determine the microtubular origin of the SH signal. For SH measurements and correlation with dimer conformation in hippocampal neurons, DMSO (Control), 10 nM taxol (4 h, Cytoskeleton Inc.), or 10 nM epothilone B (6 h, Sigma-Aldrich) was applied in plating medium before imaging. Dopaminergic neurons were treated at 7 DIV with either DMSO as control, rotenone (100 nM, Sigma-Aldrich), or a combination of rotenone (100 nM) and paclitaxel (100 nM) for 4 h before imaging microtubule alterations as an application for SH imaging. For brain slices, 10 μM taxol and 100 nM Epothilone in ACSF was used during 4 h incubation periods before recording SH signals.

**In vitro tubulin polymerization.** Tubulin protein (Cytoskeleton) was resuspended in GPEM buffer (0.2% glycerol, 80 mM Pipes-KOH; 2 mM EGTA; 1 mM MgCl2, pH 6.9) supplemented with 1 mM GTP (Cytoskeleton) and stored at −80 °C. For GTP-bound tubulin dimer HRS measurements, a vial of 1 mg ml$^{-1}$ tubulin protein was used. For GDP-bound tubulin dimer HRS measurements, tubulin protein was polymerized into microtubules by incubation for 1 h at 37 °C after which the vial was put on ice for 1 h to depolymerize the microtubules into tubulin dimers. These vials containing 1 mg ml$^{-1}$ tubulin were the first concentration after which they were diluted in PEM buffer containing GTP for the concentration series used for HRS measurements.

**Imaging.** B27 media was replaced by HEPES buffer (in mM: 148 NaCl, 5 KCl, 1 MgCl$_2$, 10 Glucose, 10 HEPES, 2 CaCl$_2$) for live cell imaging at room temperature. A Zeiss LSM 780 confocal laser scanning microscope (Zeiss) fitted with an Argon laser (488 nm) and solid state lasers (561, 633 nm) was used for fluorescence and differential interference contrast (DIC) imaging. A MaiTai DeepSee titanium sapphire tunable laser (680–1050 nm, 100 fs, Spectra-Physics) was used in combination with custom-made polarization control (based on rotatable ¼ and ½ waveplates) for SHG imaging with an incoming wavelength of 810 nm. The average power at the sample was limited to 25 mW to prevent damage to the cultures. The laser polarization direction is indicated with a double arrow or circular arrow for circular polarization. High numerical aperture (NA) water-immersion objectives (LD LCI Plan-Apochromat 25×/0.8 Imm Corr DIC M27 and LD C-Apochromat 63×/1.15 W Corr M27, Zeiss) were used in combination with either a 0.55 NA air condenser or a high (0.8) NA air condenser for recording forward SH signals. SH (emission filter: 380–430 nm) and autofluorescence (emission filter: 420–480 nm) images were simultaneously acquired with GaAsP non-descanned detectors (Zeiss BiG) in, respectively, the forward and backward detection path. All SH signals were recorded in photon counting mode and are represented as the number of photons per microsecond dwell time.

**Fixation procedures.** Cells were fixed with either 4% paraformaldehyde (PFA, 30 min, RT), 100% methanol (2 min, −20 °C), 100% acetone (10 min, −20 °C) or 5% acetic acid in ethanol (10, min RT) and washed in phosphate-buffered saline (PBS). Ethylene glycol bis(succinimidyl succinate) (EGS) fixation was performed using 2 mM EGS (Thermo Fisher Scientific) in warm GPEM buffer, followed by extensive washing in GPEM buffer, all steps at 37 °C.

**Immunofluorescence labeling.** After fixation, cells were incubated in blocking medium containing PBS, 4% serum from secondary hosts (Chemicon International), and 0.1% Triton X-100 (Sigma) (2 h, RT). Overnight incubation at 4 °C with combinations of the following primary antibodies followed: mouse HuCD (1:500, cat. number A21271, Invitrogen Molecular probes), goat TAU (1:1000, cat.

number SC1995, Santa Cruz Biotechnology), chicken MAP2 (1:5000, cat. number ab5392, Abcam), rabbit α-tubulin (1:1000, cat. number ab18251, Abcam). After washing with PBS the secondary antibodies were applied (1 h, RT). All antibodies were diluted in blocking medium. Three 10-min wash steps with PBS were performed and excessive PBS was removed. All preparations were mounted in Citifluor (Citifluor Ltd.) before imaging. For GTP-tubulin staining, we followed the protocol by Dimitrov et al.[11]. In brief, cells were treated with 0.05% Triton in GPEM buffer (3 min, 37 °C) before incubation with MB11 (1:250, cat. number AG-27B-0009-C100 Adipogen) diluted in GPEM buffer supplemented with 2% BSA (15 min, 37 °C). After a quick wash in GPEM buffer, donkey anti human alexa 594 (1:1000, cat. number 709-585-149, Jackson ImmunoResearch) was applied (15 min, 37 °C). Methanol fixation followed as described above.

**Electron microscopy**. For electron microscopy, cells were grown on glass coverslips, and fixed with 2.5% glutaraldehyde in 0.1 M cacodylate buffer (2 h, RT), followed by extensive washing of the samples in the same buffer. Then, the samples were osmicated in 2% osmiumtetroxide in 0.1 M cacodylate buffer (1 h, RT), washed and dehydrated in a graded ethanol series (5 min steps). En bloc staining with 3% uranyl acetate took place in the 70% ethanol step (30 min on ice). After washing with 100% ethanol, samples were placed in 1:1 ethanol:epoxy (30 min, on ice) and 1:2 ethanol:epoxy (O/N, RT) mixtures. The next day samples were pre-embedded in a very thin layer of epoxy. Then, regions of interest were selected that were covered with an epoxy filled inverted BEEM-capsule. Finally, samples were cured at 60 °C for 2 days. Glass coverslips were removed with hydrofluoric acid (30 min, RT) and washed twice with water. 70 nm sections were cut and post stained with uranyl acetate and lead citrate before imaging with a JEM1400 transmission electron microscope (JEOL) equipped with an EMSIS Quemesa camera (11Mpxl) at 80 kV.

**Hyper-Rayleigh scattering**. The experimental set-up used for HRS was already described elsewhere[58]. The laser system (Supplementary Fig. 6A) consisted of a mode-locked femtosecond laser (Insight DS+, Spectra-Physics) producing a vertically polarized beam with an average power output of 1.1 W at 1000 nm and a repetition rate of 80 MHz with pulse widths of 120 fs. The laser can be tuned from 680 to 1300 nm, with the output power modulated by the combination of an achromatic half-wave plate (HWP) mounted on a rotation stage and a polarizer. Mirrors (M) were used to guide the laser beam. A plano-convex lens (PCL) was used to adjust the focal point of the laser beam into the center of a 10 × 2 mm quartz cuvette containing the sample of interest. Light was collected under 90° and focused onto the entrance slit of the spectrograph (Bruker IS 500) by an achromatic, aspheric condenser lens (focal length = 30 mm). A dove prism mounted at 45° was used to obtain a 90° rotation of the image of the focal point, enabling the collection of the entire image on the slit of the spectrograph, greatly enhancing the sensitivity. A bandpass filter (FF01-790/SP-25, Semrock) was located before the entrance slit to ensure that no Rayleigh scattering could enter the spectrograph. The spectrograph (Bruker IS/SM 500) uses a grating to separate the incoming light into its different wavelength components, which were collected with an electron-multiplying charge-coupled device (EMCCD) camera (Andor Solis model iXon Ultra 897). The output power was adjusted by rotating the HWP to obtain the best signal to noise ratio while preventing any optical induced damage.

**HRS data analysis**. EMCCD images are taken for the pure solvent followed by a concentration series of the solute. The scattered light is integrated vertically over the entrance slit of the spectrograph. The HRS and two photon fluorescence (TPF) peaks are fitted with a Gaussian function using origin software. The overlap of the TPF peak and the tails before the HRS peak are used to correct for any possible multiphoton contribution to the signal and obtain the pure HRS signal[59]. For very dilute solutions, like the ones used in an HRS experiment, the refractive index of the solvent is assumed not to change. Since the hyper-Rayleigh response originates from uncorrelated scatterers (i.e., no phase relation exists between them), the total response can be considered as the sum of signals generated by all individual scatterers[60]:

$$I_{2\omega} = \frac{16\pi^5}{c\lambda^4 r^2} N f_\omega^4 f_{2\omega}^2 \langle \beta_{\text{HRS}}^2 \rangle I_\omega^2 \tag{1}$$

with $N$ being the number density of molecular scatters, $\lambda$ the fundamental wavelength, $c$ the speed of light, $I_\omega$ the fundamental laser intensity, $I_{2\omega}$ the intensity of the frequency-doubled light and $f_\omega$ and $f_{2\omega}$ local field factors that account for the effect of induced dipoles in the medium through electronic polarization. The brackets around $\beta$ indicate orientational averaging. For a two-component system, consisting of solvent (S) and randomly oriented tubulin (tub), the following equations hold for both the sample and reference compound:

$$I_{2\omega,x} = g\left(N_S\langle\beta_{zzz}^2\rangle_S + N_{tub}\langle\beta_{zzz}^2\rangle_{tub}\right)I_\omega^2 \tag{2}$$

$$I_{2\omega,\text{ref}} = g\left(N_S\langle\beta_{zzz}^2\rangle_S + N_{\text{ref}}\langle\beta_{zzz}^2\rangle_{\text{ref}}\right)I_\omega^2 \tag{3}$$

with $I_{2\omega,x}$ and $I_{2\omega,ref}$ being the frequency-doubled intensity for the compound of

interest and of the reference compound, respectively. $g$ is a constant accounting for the local field factors, instrumental parameters, and the scattering geometry. If we measure a concentration series of both the sample and the reference, we can plot the area of the integrated HRS peak obtained from each spectrum versus the concentration:

$$I_{2\omega} = N_S\left(g\langle\beta_{zzz}^2\rangle I_\omega^2\right) + N_{tub}\left(g\langle\beta_{zzz}^2\rangle I_\omega^2\right) \tag{4}$$

with $N_S$ and $N_{tub}$ being the number densities of solvent and solute, respectively. For the internal reference method (IRM), where we calibrate to the solvent itself, we can rewrite Eq. (4) as:

$$\frac{I_{2\omega}}{I_\omega^2} = g\left(N_S\langle\beta_{zzz}^2\rangle_S + N_{tub}\langle\beta_{zzz}^2\rangle_{tub}\right) \tag{5}$$

This equation has the form of y = a + N_{tub}b with the intercept $a = gN_S\langle\beta_{zzz}^2\rangle_S$ and slope $b = g\langle\beta_{zzz}^2\rangle_{tub}$ in order to get:

$$\sqrt{\langle\beta_{zzz}^2\rangle_{tub}} = \sqrt{\frac{\text{Slope}}{\text{Intercept}}} N_S\langle\beta_{zzz}^2\rangle_S \tag{6}$$

with $\sqrt{\beta_{zzz}^2}$ the hyperpolarizability tensor of the solvent and $N_S$ its number density (M). In this work, we use the static hyperpolarizability data and transition wavelengths of the solvent from Campo et al.[59]. To calculate the $\beta_{zzz}$ at a certain wavelength we use the two level model (TLM) proposed by Oudar et al.[61].

$$\beta_{zzz} = \frac{\beta_{zzz,0}}{\left(1 - \left(\frac{\lambda_{eg}}{\omega}\right)^2\right)\left(1 - \left(\frac{\lambda_{eg}}{2\omega}\right)^2\right)} \tag{7}$$

where $\beta_{zzz,0}$ is the static hyperpolarizability and $\lambda_{eg}$ the transition wavelength of the solvent molecule. The $\beta_{zzz,0}$ for water is $0.186 \times 10^{-30}$ esu with $\lambda_{eg}$ being 146 nm. Depolarization ratios (intensity ratio for vertically scattered hyper-Rayleigh intensity over horizontally scattered intensity for vertical input laser polarization) were also determined to estimate the importance of the dipolar $\beta_{zzz}$ tensor element with respect to other contributing tensor elements[32].

**Analysis and statistics**. Intensity analysis was performed using Fiji. For SH imaging, fibers were traced and average intensity along the fiber calculated. For circular polarization, all fiber directions were included. For horizontal polarization only horizontal fibers were included. In-house Igor code was used to generate kymographs and analyze EB3-eGFP time-lapse recordings. Graphpad was used for statistical analysis: $p < 0.05$ (*), $p < 0.01$ (**), $p < 0.001$ (***) and bar graphs represent mean values with standard error of the mean. N-values represent the number of cells from three separate cultures. Shapiro–Wilk normality tests were used to assess the normal distribution of the data. T-tests were conducted as a two-sided test.

## Data availability
The raw recordings that support the findings of this study are available from the corresponding author upon reasonable request. The source data underlying all main and supplementary figures are provided as a Source Data File.

## Code availability
Custom-written Igor code is available from the corresponding author upon reasonable request.

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

## Acknowledgements

We would like to thank Michael Moons for technical support, Tobias Martens for analyzing SH recordings of brain slices, Michiel Martens, Youcef Kazwiny, and all current and past LENS members for scientific comments on the project and help with preliminary data collection and set-up refinement. The authors' work is supported by the Research Foundation Flanders (FWO) grant G.0929.15 (to P.V.B., K.C., M.A.) and Hercules AKUL/11/37 (to P.V.B.).

## Author contributions

Methodology: V.V.S., P.V.B. Investigation: V.V.S., K.V., P.B., Z.L., Y.d.C. Data discussion: V.V.S., W.B., I.D., M.A., K.C., P.V.B. Resources: I.D., W.B., K.C., P.V.B. Writing—original draft: V.V.S. Writing—review and editing: V.V.S., W.B., Y.d.C., I.D., M.A., K.C., P.V.B.. Conceptualization: V.V.S., P.V.B. Supervision: P.V.B. Funding acquisition: M.A., K.C., P.V.B.

## Additional information

**Competing interests:** The authors declare no competing interests.

