## [Transparent Peer Review File · Nature Communications]

Reviewers' comments:

Reviewer #1 (Remarks to the Author):

In this manuscript, Steenbergen et al use a combination of second harmonic (SH) imaging microscopy with fluorescence microscopy to investigate the molecular basis for how microtubules generate SH signals. While the specific question of what properties of microtubules yield SH signals could be very useful and of substantial interest to the cytoskeleton and neuronal cell biology field, I have the following major concerns and some minor comments about the authors' conclusions as stated in the manuscript in its current form.

Major comments:

1. Based on their data in Figures 4 and 5, the authors state that 'GTP-bound tubulin dimers are indispensable for SHG'. Given that the data in these figures suggests a correlation between GTP-bound tubulin and detectable SHG signal, I think this is a particularly strong statement:

a. In neurons cultured from Tau deficient mice, the authors observe significantly less SHG signal compared to wildtype mice. If the authors can demonstrate that lack of Tau causes a complete ablation of GTP-bound tubulin, it is then conceivable that GTP-bound tubulin is absolutely essential for detectable SHG signal. However, if anything, Tau is known to promote the polymerization rate of microtubules and I am not aware of any study showing deletion of Tau altering GTP-tubulin state. Given the innumerable interactors of microtubules and several properties of microtubules, it is likely that multiple factors contribute to SHG signal and GTP-tubulin is very likely one of them.

b. If the GTP-bound tubulin alone is a predominant factor for SHG signal, I would expect a much stronger correlation between these two. In Figure 5A, the authors report a Pearson correlation less than 0.5 which to me indicates this is not the only determining factor at play.

c. In figure S3, the authors use immunofluorescence images to estimate the amount of GTP-tubulin in control vs taxol treated neurons. They report a statistically significant increase of about 10% in GTP-tubulin in taxol treated neurons. Given the long and elaborate morphology of a neuron, I am not sure that this increase is biologically significant. Even if this is a biologically meaningful increase that could impact several processes in neurons, I find the stark difference in SHG signal in the images of control vs taxol treated neurons shown in the figure intriguing, given the generally small increase in GTP-tubulin. This is confusing in light of the correlation the authors report (point 1b) and also raises concern about using taxol in these experiments (see point 2)

2. In Figure S4, the authors report data that taxol treated HEK293T cells have detectable SHG signal but these same cells when fixed after taxol treatment no longer have observable SHG signal. It would be useful to do a similar experiment in neurons as well, as the microtubule dynamics and the local environment of axons and dendrites are very different from those of HEK cells.

a. Additionally, can the authors use other microtubule stabilizing drugs to make this point about GTP-tubulin correlating with SHG signal. Bradke and colleagues (Ruschel et al 2015) showed that epothilone B is a microtubule stabilizing molecule.

b. Would using a more common drug like nocodazole, at low doses cause the same effect as taxol with SHG signal detection?

3. Finally, can the authors perform these experiments in brain slices? Specifically showing (i) staining of GTP-tubulin correlating with SHG signal and (ii) application of taxol increasing SHG signal. These ex vivo data would not only complement their in vitro findings very well, they will also demonstrate the broad applicability of SHG microscopy for neuroscientists tackling questions

of microtubules in vivo.

In light of these specific comments, I think the reader would benefit greatly if the authors provided more lines of evidence to support their claim on GTP-tubulin being the most important factor in SHG signal generation by microtubules.

Minor comments:

1. As SHG signals can sometimes be hard to see in print, I strongly encourage the authors to consider showing all their images in grayscale.
2. All figures showing neurons should mention the age of the neuronal culture either in the Figure legend or the figure itself. This is a very important factor in this study as it is well known that microtubule polarity changes with time in a developing neuron in culture.
3. In Figure 1G, why does the midbrain neuron have SHG signal in most of its neurites? Is this just a neuron with multiple axons? Or is this a case where even dendrites have detectable SHG signal here?
4. In Figure 2B, why is EB3 localized to the axon alone with no signal in dendrites? Since EB3 labels microtubule tips, it is important to select neurons expressing low-levels of this protein so only the plus tips are labeled by EB3. However, in this image, EB3 seems to be expressed all over and is predominantly cytoplasmic.
5. What is the age of the neuron in Figure 2B? This becomes important for the reader especially given their data in Figure 6C.

Reviewer #2 (Remarks to the Author):

In this paper, the authors investigate the parameters of SHG in microtubules. In agreement with previously published results, they show that microtubules are responsible for SHG in live neurons, and that axonal microtubules are much more prone to SHG than dendritic ones. Several novel findings are reported.

- Using EB1-GFP labelling and kymographs of axonal microtubule growth, the authors detect oriented microtubule bundles that fail to generate SH signals, indicating that the uniform polarity of axonal microtubules (contrary to the mixed orientations of their dendritic counterparts) is not sufficient to explain their ability to produce the SHG signal.
- The work identifies the cross-linking agent EGS as a fixative that preserves SHG, unlike usual fixatives such as PFA or methanol.
- The authors then use EGS fixation and immunostaining with the conformation-sensitive mab MB11 to identify a positive correlation between MB11 staining and SHG strength. They posit that the conformation change responsible for MB11 binding arises from the presence of GTP-loaded tubulin in the microtubule lattice. This latter makes microtubules more rigid.
- Using measurements of hyper-Rayleigh scattering with purified tubulin dimers, the authors show that GTP-tubulin has two-fold larger dipolar hyperpolarizability than GDP-tubulin. Along with the effect of parallel, rigid bundling and ensuing dipole alignment, incorporation of GTP-tubulin would hence be expected to increase SHG by microtubules.
- Straightening microtubule bundles with taxol is also shown to increase SHG.

Based on these findings, the authors propose that SHG in neurons arises from microtubule bundles in a straight, parallel conformation containing GTP-tubulin. The authors also show that in live neurons, SHG responds to disruption of axonal microtubules by rotenone, restoration by taxol, and conformational changes induced by KO of the microtubule-associated Tau protein.

This paper brings useful new information regarding the generation of SH signals in microtubules. The biological data look convincing, even though the immunostaining and GFP images are not of the highest quality. However, the proposal that SHG changes are related to transitions from GTP to

GDP tubulin is not new. Psilodimitrakopoulos et al. (2013; duly cited in the present manuscript) discussed the idea based on a geometric interpretation of anisotropy measurements. While the present data explore this further, that the MB11 antibody specifically targets GTP-tubulin is controversial in the field; and GTP-tubulin has so far only been proven to exist at the +end of microtubules. Indeed the epitope recognized by MB11 antibody is questionable as indicated in the last publication of Perez and Poüs (de Forges H, Pilon A, Cantaloube I, Pallandre A, Haghiri-Gosnet AM, Perez F, Poüs C. *Curr Biol.* 2016 Dec 19;26(24):3399-3406) where the authors used the words "previously termed "GTP islands". MB11 antibody is thought to potentially recognize some "defects" on microtubule lattice (holes, number of protofilament transition zone, etc..). The discrepancies between axons and dendrites in SHG signal can be linked to differential number of lattice defects (resulting of difference on microtubule growth speed for example). Thus to our point of view, it remains unclear how differences in GTP-tubulin may account for the presence or absence of SHG along entire stretches of microtubule length. The statement "identification the GTP-bound tubulin dimer conformation as the main origin of the SH signal" which is the main message of the work, is thus somehow questionable.

Reviewer #3 (Remarks to the Author):

This manuscript by Van Stenbergen et al describes a study of the molecular origin of second harmonic generation (SHG) in tubulin. The main original finding is that GTP-bound tubulin dimer conformation is necessary for microtubules to generate detectable SHG signals. This point is properly established in the study by an extensive combination of complementary measurements involving in vitro / in vivo SHG imaging, electron microscopy, hyper-Rayleigh scattering measurements, and chemical cell treatments. Its consequence for biological studies is briefly illustrated in well-chosen contexts. This work has important implications for the development & understanding of label-free SHG microscopy of tubulin structures.

Overall, this is a important ground study on the topic of SHG imaging of tubulin, which advances the field. The quality and significance of this work is superior to that of the majority of recent publications discussing progress in SHG imaging, and it without any doubt deserves publication in a good journal.

Suitability for Nature Communications is left to the Editor's appreciation. This work will clearly be of importance to scientists working on nonlinear microscopy developments; it may appeal to a broader audience, but this will probably depend on the future use of SHG imaging in tubulin studies.

Below are some major and minor issues that should be addressed before publication.

Comments on the content

1) It would be useful to provide a quantification of SHG signal levels. The authors specify that they use up to 25 mW of excitation power (680-1050nm, 100fs) focused with a ~1NA objective. They should also give an estimate of typical detected SHG fluxes (detected photons/ μs), or at least of the signal-to-noise ratio of the tubulin structures in the images.

This quantification is important, because tubulin-SHG signals are significantly weaker than collagen-SHG or myofilament-SHG, which may limit their usefulness for in vivo studies.

2) Likewise, the authors write in the abstract and in the text that "The use of SH imaging of microtubules remains limited, mostly because the true molecular origin and main determinants required to generate SH from microtubules are not fully understood."

This statement should be tempered. One at least equally important reason for the limited use of tubulin-SHG signals is that these signals are generally weak, and difficult to use in vivo without inducing cell damage.

3) Page 9 "the microtubule SH signal intensity measured in living cells positively correlated with fluorescence intensities of an α -tubulin immunostaining"

A simple model would predict that, because of the coherence, SHG is expected to scale as the square of the fluorescence signal, in the case of tubulin structures smaller than the axial resolution.

The authors could discuss whether their data are consistent with such a straightforward analysis (however, the low signal-to-noise ratio may prevent such a quantitative discussion).

4) Page 9 "The cellular process with low SHG signals clearly contained less parallel organized microtubules as compared to the parallel-aligned microtubules found in the intense SH generating process"

The authors should provide an estimation of the microtubules density based on the EM images, rather than stating "clearly contained". A simple prediction of SHG signal level from this density would also be interesting.

Comments on the presentation

Page 4 – define "SHIM", or use "SHG imaging" instead.

Page 7 – "In dendrites, microtubules are not uniformly polarized leading to destructive interference"

-> leading to *partial* destructive interference (suggestion)

Page 7 "unidirectional movement of EB3-eGFP comets (Fig. 2C)" -> Fig. 2D

In the introduction about SHG imaging of microtubules, the authors write "SHG (...) can also be used to detect the far weaker tubulin dimer as it organizes itself in the microtubule lattice (Campagnola 2001, Stoothoff 2008, Psilodimitrakopoulos 2009).

-> Other studies should probably be mentioned here, e.g. Dombeck PNAS 2003, Chi-Kuang Sun J.Struct.Biol. 2004, Olivier Science 2010.

Along the same line, page 17-18, the authors write "we found that, apart from axons, also non-neuronal cells such as fibroblasts are capable of generating SH signals from their microtubules provided that sufficient detection sensitivity is available"

-> this is not a "finding", since it was used in previous studies such as Campagnola BJ 2002, Sun JSB 2004, Olivier Science 2010.

Dear Reviewers,

Thank you for your constructive comments and questions, which have helped to improve our manuscript. We hope that the changes and additional experiments in the revised manuscript address your comments and make it acceptable for publication in Nature communications.

Sincerely,

Pieter Vanden Berghe

Reviewer #1 (Remarks to the Author):

1. Based on their data in Figures 4 and 5, the authors state that ‘GTP-bound tubulin dimers are indispensable for SHG’. Given that the data in these figures suggests a correlation between GTP-bound tubulin and detectable SHG signal, I think this is a particularly strong statement:

a. In neurons cultured from Tau deficient mice, the authors observe significantly less SHG signal compared to wildtype mice. If the authors can demonstrate that lack of Tau causes a complete ablation of GTP-bound tubulin, it is then conceivable that GTP-bound tubulin is absolutely essential for detectable SHG signal.

b. If the GTP-bound tubulin alone is a predominant factor for SHG signal, I would expect a much stronger correlation between these two.

c. I find the stark difference in SHG signal in the images of control vs taxol treated neurons shown in the figure intriguing, given the generally small increase in GTP-tubulin. This is confusing in light of the correlation the authors report (point 1b) and also raises concern about using taxol in these experiments (see point 2)

We agree with the reviewer that the conformational state of tubulin by itself is not sufficient to explain all of the SH signal intensity in biological samples. We apologize that this was not clear from our original text, which we have now amended to make our conclusions more clear.

We have also changed the statement about the fact that GTP would be indispensable. It now reads: “The elevated hyperpolarizability of the GTP-tubulin dimer, makes this conformation a very important and strong contributor to the overall SH signal from microtubules”, which is based on the improved correlation between SHG signal intensities and GTP-tubulin (but less so for α -tubulin) fluorescence intensity. The theoretically expected power relation that exists between the number of sources and the SH signal itself (see below and comments to reviewer 3) also confirms the role of GTP-tubulin in generating SH.

The importance of the GTP conformation is further corroborated by the increased hyperpolarisability tensor of GTP-bound tubulin as detected using Hyper-Rayleigh scattering. Because of the latter, we know now that at the molecular level, GTP-bound tubulin dimers will generate much stronger SH signals than do GDP molecules. The GTP dependency of the SH signals, is further underscored by the fact that SH signals from microtubules are not retained upon fixation with common fixatives such as PFA, as this specific conformation is

lost during the fixation process. However, the signal generated at the molecular level is not solely responsible for the overall detected SH signal in biological samples. Here, many dimers are incorporated in numerous microtubule arrays. SHG in biological samples is therefore dependent on multiple other factors apart from the molecular conformation, such as microtubule number, polarization direction and organization of the microtubule network as shown in the manuscript.

a/ We agree with the reviewer that we have no proof that Tau knock-out mice have less GTP-bound tubulin dimers compared with wild-type animals. We definitely did not intend to make that point. Also in our own staining (Supplementary figure 6) we did not find a significant difference in total tubulin or GTP-bound tubulin levels. The absence of Tau reduces the overall signal in another way. As Tau regulates inter-microtubule spacing and overall microtubule network organisation, these results nicely indicate that also the other contributing factors such as microtubule organisation are of importance to generate a detectable signal in biological samples.

b/ While the presence of a GTP-bound tubulin dimer conformation is the main factor that determines the capacity to generate SH signals at the molecular level, also the other elements (microtubule number, polarity and organization) are important in biological samples with multiple microtubule arrays. These factors influence the overall detected SH signal intensity and add noise to the correlation and thus a less stringent correlation between GTP-tubulin dimer conformation and SHG signal intensity is obtained. Furthermore it needs to be noted that uncertainty is also introduced during the measurements as SH is measured while the neurons are alive, after which the GTP specific immunolabeling is performed in triton permeabilized (but not fixed) neurons at 37⁰ C. Despite these biological and technical confounders the expected power relation between number of sources and SH intensity is still detectable. We now show (Figure 5A) the relation between GTP-tubulin and SHG with a Spearman correlation coefficient of 0.5642.

Although the data points are fairly scattered, an increased Spearman correlation was found between SH and MB11 ($r = 0.56$) compared with α - tubulin (0.29). Furthermore the power fit (red line) through the MB11 data (0.345) is closer to the theoretical 0.5 power compared compared to the α -tubulin data (0.10). Because of technical (signal to noise) and biological uncertainty we chose not to include these power fits in the paper. However if the reviewer deems this necessary we can definitely do so.

c/ The distinct increase in signal intensities upon addition of taxol for the GTP-tubulin signal intensity (35% CTRL versus Taxol) and SHG signal intensity (86% CTRL versus Taxol) results from the non-linear character of SHG. The detected SHG signal intensity scales as the square of the fluorescence signal. This explains why upon addition of taxol, a more pronounced difference is observed in the SH signal as compared to the GTP-tubulin staining.

2. In Figure S4, the authors report data that taxol treated HEK293T cells have detectable SHG signal but these same cells when fixed after taxol treatment no longer have observable SHG signal. It would be useful to do a similar experiment in neurons as well.

We thank the reviewer for bringing this to our attention; this is indeed a very useful suggestion since almost all experiments were performed using neuronal cultures. We have added a panel to illustrate the effect of (PFA) fixation in neuronal cultures incubated with taxol (Supplementary Figure 4). Similarly as in HEK cells, the SH signal in neurons is lost upon fixation.

a. Additionally, can the authors use other microtubule stabilizing drugs to make this point about GTP-tubulin correlating with SHG signal. Bradke and colleagues (Ruschel et al 2015) showed that epothilone B is a microtubule stabilizing molecule.

As suggested by the reviewer, we have performed an extra set of experiments to show the action of epothilone B. Using a concentration of 10 nM for 6 hours, we found indeed a significant increase in SH signal intensity compared to control cells. As these results nicely confirm the role of MT stabilizing drugs, we have included them as an extra panel in figure 5.

b. Would using a more common drug like nocodazole, at low doses cause the same effect as taxol with SHG signal detection?

As nocodazole inhibits polymerization of new microtubules, we would expect it to reduce the microtubule number and hence result in decreased SH signal intensities. We performed experiments in which we treated the neurons with a low concentration of nocodazole (50 nM) for 4 hours. We found a small but statistically significant decrease in SH signal intensity compared to controls. As nocodazole specifically targets microtubules, the resulting change in SH signal intensities corroborates the data presented in the first results section, in which we confirm the microtubular origin of the detected signal. Therefore, we included this data as an extra panel F in figure 1.

3. Finally, can the authors perform these experiments in brain slices? Specifically showing (i) staining of GTP-tubulin correlating with SHG signal and (ii) application of taxol increasing SHG signal. These ex vivo data would not only complement their in vitro findings very well, they will also demonstrate the broad applicability of SHG microscopy for neuroscientists tackling questions of microtubules in vivo.

We would like to thank the reviewer for this question, as we agree that the use of SH imaging in brain slices will be useful for the neuroscience field.

We are afraid that due to a number of technical reasons the suggested immunostainings in a thick brain slice are not possible. First, since the signal disappears upon fixation, we have to cut the slices from fresh brains, which prevents us from cutting slices thinner than 250 µm. Immunostainings are not trivial in these thick slices because antibodies do not penetrate easily and the high density of neuronal fibers leads to low signal to noise. The use of the MB11 antibody to recognize GTP-tubulin dimers further complicated the procedure, as cells remain alive while applying triton to allow the antibodies to penetrate and the entire staining process has to be concluded within 30 minutes before fixing the cells. These short incubation times are not sufficient to reach the deeper cell layers inside the slices, where we detect SH signals. We have tried to perform this staining, combined with an α-tubulin antibody in EGS fixed brain slices to allow increased incubation times but failed to detect individual fibers in either channel, due to the afore mentioned technical constraints. To our knowledge, for any brain slice stainings that have been reported, thin cryo-sections were used that were fixed

using PFA before snap-freezing and thin sectioning into 10-50 μm layers. As the process to obtain cryo-sections leads to disappearance of the signal, we were presently not able to perform the requested experiment.

ii/ Unlike the antibody labeling, live imaging of drug effects in brain slices is possible. Below you can find an overview tile scan recording of SH signals in an acute hippocampal slice. Sc Subiculum, DG Dendate Gyrus, CA1-3: Cornu Ammonis.

We followed the suggestion of the reviewer and used acute hippocampal brain slices to test the effects of taxol and epothilone B (DMSO was used as control). We opted for the CA1 layer of the hippocampus as here, neuronal cell bodies are highly organized and their projections run fairly parallel for hundreds of micrometer, making it easier to localize the correct region for comparison between conditions. We found a significant increase in SH signal intensities in both taxol (10 μM , 4 hours) and epothilone (100 nM, 4 hours) treated slices compared to controls. As we agree with the reviewer's suggestion that these experiments will be beneficial to demonstrate the broad usability of SH imaging of microtubules in the neuroscience field, we included these results as a new panel D in figure 6.

Minor comments:

1. As SHG signals can sometimes be hard to see in print, I strongly encourage the authors to consider showing all their images in grayscale.

We understand that some printed versions may not be very clear compared to the digital version. As we also combine the SH signal with other channels such as autofluorescence, we are not convinced that the SH signal in grayscale in combination with another color for autofluorescence would make the images more clear. We have regenerated all the figures in corelDRAW to be vector based which improved overall picture quality and allows on-screen magnification without loss of detail. We hope that the reviewer agrees that in this version the image quality is sufficient to judge SH signals correctly.

2. All figures showing neurons should mention the age of the neuronal culture either in the Figure legend or the figure itself. This is a very important factor in this study as it is well known that microtubule polarity changes with time in a developing neuron in culture.

As mentioned in the methods section, all neuronal cultures were imaged at 7 days *in vitro* (DIV) unless specifically mentioned otherwise. We agree that it is easier to have this information close to the figure as well, therefore we added the neuronal ages in each of the figure legends.

3. In Figure 1G, why does the midbrain neuron have SHG signal in most of its neurites? Is this just a neuron with multiple axons? Or is this a case where even dendrites have detectable SHG signal here?

This is indeed a case where SH signals are also detected in dendrites. With sufficient detection sensitivity SH recordings can also be made from the dendritic microtubule network, if non-centrosymmetric requirements are fulfilled. However, in most cases dendrites have mixed microtubule polarity and therefore due to symmetric reasons cancels the possibility to generate SH.

4. In Figure 2B, why is EB3 localized to the axon alone with no signal in dendrites? Since EB3 label microtubule tips, it is important to select neurons expressing low-levels of this protein so only the plus tips are labeled by EB3. However, in this image, EB3 seems to be expressed all over and is predominantly cytoplasmic.

In the specific neuron outlined in the previous manuscript version, EB3 was also localized to the dendritic compartment, but as the dendrites were rather short it was hard to make this visible in a large overview figure. We have repeated the EB3-SHG experiments with a lower concentration of EB3 (0.5 μ g per coverslip) as suggested and also proceeded with an axonal and dendritic staining after live EB3 and SHG recordings. This allowed us to detect and quantify isolated EB3 comets in both types of processes. We would like to emphasize that, because of the combination with SH imaging, these recordings were performed on a confocal microscope, and not using TIRF microscopy. As a consequence, our z-resolution is not comparable to TIRF-based EB3 studies and therefore EB3 expression appears less spotted and more cytoplasmic. However, the confocal recordings have in this context an advantage, in that they capture all EB3 events that happen in the processes, and not only the ones at the membrane (TIRF), and are as such, more representative of the orientation of all MT in the axon and dendrites.

Our findings, that EB3 comets are 80% unidirectional in axons and significantly less unidirectional in dendrites furthermore show that our EB3 measurements are indeed correct as they correspond well with microtubule polarity reported in literature. We have changed figure 2 to include the new data.

5. What is the age of the neuron in Figure 2B? This becomes important for the reader especially given their data in Figure 6C.

These neurons were recorded at 7 DIV. We understand the reviewer's question as intense SH signals at this age would be mainly localized at the neuronal terminal. We would like to note that in these cultures, it is not straight-forward to include both the cell soma and growth cone in one single field of view such as for figure 6C. Therefore, the neuron seen in the previous version of figure 2 might not seem to have high SH signal intensities at end of the axonal projection, but in fact the end of the fiber is not visible in the image. We have included the previous neuron's EB3-GFP maximum projection below with increased brightness of the figure, where you can see that the fiber is still continuing outside the recorded frame (arrows). As we need sufficient temporal resolution to accurately track EB3 comets, we are not able to record tile scans to include the entire fiber for this specific experiment.

Reviewer #2 (Remarks to the Author):

In this paper, the authors investigate the parameters of SHG in microtubules. In agreement with previously published results, they show that microtubules are responsible for SHG in live neurons, and that axonal microtubules are much more prone to SHG than dendritic ones. Several novel findings are reported.

- Using EB1-GFP labelling and kymographs of axonal microtubule growth, the authors detect oriented microtubule bundles that fail to generate SH signals, indicating that the uniform polarity of axonal microtubules (contrary to the mixed orientations of their dendritic counterparts) is not sufficient to explain their ability to produce the SHG signal.
- The work identifies the cross-linking agent EGS as a fixative that preserves SHG, unlike usual fixatives such as PFA or methanol.
- The authors then use EGS fixation and immunostaining with the conformation-sensitive mab MB11 to identify a positive correlation between MB11 staining and SHG strength. They posit that the conformation change responsible for MB11 binding arises from the presence of GTP-loaded tubulin in the microtubule lattice. This latter makes microtubules more rigid.
- Using measurements of hyper-Rayleigh scattering with purified tubulin dimers, the authors show that GTP-tubulin has two-fold larger dipolar hyperpolarizability than GDP-tubulin. Along with the effect of parallel, rigid bundling and ensuing dipole alignment, incorporation of GTP-tubulin would hence be expected to increase SHG by microtubules.
- Straightening microtubule bundles with taxol is also shown to increase SHG. Based on these findings, the authors propose that SHG in neurons arises from microtubule bundles in a straight, parallel conformation containing GTP-tubulin. The authors also show that in live neurons, SHG responds to disruption of axonal microtubules by rotenone, restoration by taxol, and conformational changes induced by KO of the microtubule-associated Tau protein.

This paper brings useful new information regarding the generation of SH signals in microtubules. The biological data look convincing, even though the immunostaining and GFP images are not of the highest quality. However, the proposal that SHG changes are related to transitions from GTP to GDP tubulin is not new. Psilodimitrakopoulos et al. (2013; duly cited in the present manuscript) discussed the idea based on a geometric interpretation of anisotropy measurements.

While the present data explore this further, that the MB11 antibody specifically targets GTP-tubulin is controversial in the field; and GTP-tubulin has so far only been proven to exist at the +end of microtubules. Indeed the epitope recognized by MB11 antibody is questionable as indicated in the last publication of Perez and Poüs (de Forges H, Pilon A, Cantaloube I, Pallandre A, Haghiri-Gosnet AM, Perez F, Poüs C. *Curr Biol.* 2016 Dec 19;26(24):3399-3406) where the authors used the words “previously termed "GTP islands". MB11 antibody is thought to potentially recognize some “defects” on microtubule lattice (holes, number of protofilament transition zone, etc..). The discrepancies between axons and dendrites in SHG signal can be linked to differential number of lattice defects (resulting of difference on microtubule growth speed for example). Thus to our point of view, it remains unclear how differences in GTP-tubulin may account for the presence or absence of SHG along entire stretches of microtubule length. The statement “identification the GTP-bound tubulin dimer conformation as the main origin of the SH signal” which is the main message of the work, is thus somehow questionable.

Thank you very much for the positive appreciation of our work. We also thank the reviewer for raising the comments related to GTP tubulin, which we have addressed both experimentally and by adding references to recently published work. As mentioned in our first submission, and in the comments of the reviewer, the possible involvement of the GTP conformation of tubulin was suggested based on anisotropy measurements (*Psilodimitrakoupolos, et al 2013*). In this paper, we provide experimental evidence for the important role of the GTP conformation in SH generation. To do this, we used several different approaches as none of them are perfectly suited to be applied in a biological setting. The reviewer correctly raises the point that the antibody used in this study, MB11 directed against GTP-tubulin has been shown to preferentially bind to regions in the microtubule lattice where defects have occurred (*de Forges, Curr Biol 2016*). Though not proven, it may well be that these damaged sites do contain GTP-bound tubulin dimers. A study by Vemu et al (*Science, 2018*) and Aumeier et al (*Nature Cell Biology, 2016*) for instance indicate that defects in the microtubule lattice are repaired by introducing GTP-bound tubulin dimers, which re-stabilize the microtubule and protect it from depolymerizing. Also in the study mentioned by the reviewer, it is possible that sites detected are the ones where damage has occurred and have been repaired by introducing new GTP-bound tubulin dimers during the polymerization process. However, the conclusion that GTP tubulin is the most important contributor to the SHG signal at the molecular level, is not only based on the GTP tubulin specific antibody staining. The Hyper-Rayleigh scattering experiments on both GTP- and GDP- bound tubulin dimers show that GTP-bound dimers have a larger hyperpolarizability tensor, and are thus more efficient SH generators as compared to GDP-bound tubulin dimers. The fact that the GTP conformation is not maintained during fixation aligns well with the fact that MT based SH is absent in fixed cells. Together we believe that this is a strong argument for GTP-tubulin as the main contributor in SH generation at the molecular level.

Still, we understand the reviewer's question about the apparent continuity of the SH signal. Indeed, how these discrete GTP-bound tubulin dimers puncta along microtubules can lead to entire stretches of SH signals in our neuronal fibers, is intriguing but not too difficult to envision in case multiple GTP-tubulin sites are present. Most studies looking into the presence and role of GTP-bound tubulin dimers have been performed in cell lines where individual microtubules in the microtubule network are easily resolved. However, when microtubules are confined in space, as for instance in the fibroblast included in the adjusted supplementary figure 2 (A), these GTP-sites along microtubules are densely packed, leading to seemingly long stretches of GTP-tubulin (here labeled with GTP-tubulin antibody). The dense packing and parallel alignment of these GTP-tubulin sites in these regions allow for the detection of SH signals. We included a schematic in suppl fig 2 A to clarify this point.

In axonal projections, reaching an average diameter of 1 micrometer, resolving individual microtubules becomes impossible for both diffraction-limited and even some of the more advanced super-resolution microscopy techniques. The same holds true for SHG signals. Although these are generated at the single dimer level, the detectable SH signal arises from constructive interference of multiple (~ 2000 , based on regularly spaced 13-protofilament microtubules, 200 nm lateral resolution and 600 nm axial resolution) GTP-dimers within the excitation volume. Nakata et al (*J. Cell Biology 2011*) have been successful in using the GTP-bound tubulin antibody in electron micrographs, which allowed them to visualize the GTP-tubulin as individual puncta within a neuronal projection. These images suggest the presence of a lot of GTP-bound tubulin dimers to be located within 200 nm distance of each other, too close to resolve using confocal or two-photon microscopy.

Figure taken from Nakata et al ,*J. Cell Biology 2011*. Double-label immunoelectron microscopy of axons with EB1 (10 nm) and hMB11 (5 nm).

Reviewer #3 (Remarks to the Author):

This manuscript by Van Stenbergen et al describes a study of the molecular origin of second harmonic generation (SHG) in tubulin. The main original finding is that GTP-bound tubulin dimer conformation is necessary for microtubules to generate detectable SHG signals. This point is properly established in the study by an extensive combination of complementary measurements involving in vitro / in vivo SHG imaging, electron microscopy, hyper-Rayleigh scattering measurements, and chemical cell treatments. Its consequence for biological studies is briefly illustrated in well-chosen contexts. This work has important implications for the development & understanding of label-free SHG microscopy of tubulin structures.

Overall, this is a important ground study on the topic of SHG imaging of tubulin, which advances the field. The quality and significance of this work is superior to that of the majority of recent publications discussing progress in SHG imaging, and it without any doubt deserves publication in a good journal. Suitability for Nature Communications is left to the Editor's appreciation. This work will clearly be of importance to scientists working on nonlinear microscopy developments; it may appeal to a broader audience, but this will probably depend on the future use of SHG imaging in tubulin studies.

Below are some major and minor issues that should be addressed before publication.

We would like to thank the reviewer for their constructive comments and questions. Please find below a point-to-point response to the issues raised.

Comments on the content

1) It would be useful to provide a quantification of SHG signal levels. The authors specify that they use up to 25 mW of excitation power (680-1050nm, 100fs) focused with a ~1NA objective. They should also give an estimate of typical detected SHG fluxes (detected photons/ μ s), or at least of the signal-to-noise ratio of the tubulin structures in the images. This quantification is important, because tubulin-SHG signals are significantly weaker than collagen-SHG or myofilament-SHG, which may limit their usefulness for in vivo studies.

This is indeed an important point, we agree that the MT generated SH signals are definitely not as strong as those generated by other biomolecules like for instance collagen. To allow later comparisons, quantification of the numbers of photons generated is indeed very informative. We have adjusted all SHG intensities in the manuscript to be represented as SHG photons per microsecond dwell time.

2) Likewise, the authors write in the abstract and in the text that "The use of SH imaging of microtubules remains limited, mostly because the true molecular origin and main determinants required to generate SH from microtubules are not fully understood." This statement should be tempered. One at least equally important reason for the limited use of tubulin-SHG signals is that these signals are generally weak, and difficult to use in vivo without inducing cell damage.

We agree with the reviewer that detecting SH signals from microtubules is not as straight forward as for instance from myosin or collagen, the two-photon system requires sensitive detection to minimize incident optical powers and prevent damage. Sensitive detection and equally important healthy biological samples are both crucial to allow SH recordings from microtubules. We have adjusted the manuscript's abstract and introduction accordingly.

3) Page 9 “the microtubule SH signal intensity measured in living cells positively correlated with fluorescence intensities of an α -tubulin immunostaining”
 A simple model would predict that, because of the coherence, SHG is expected to scale as the square of the fluorescence signal, in the case of tubulin structures smaller than the axial resolution.

The authors could discuss whether their data are consistent with such a straightforward analysis (however, the low signal-to-noise ratio may prevent such a quantitative discussion).

This is indeed an interesting suggestion; based on theory, the SH signal should indeed scale with the square of the number of emitters. We have plotted the SHG data once in relation to the α -tubulin staining, which represents the overall microtubule density (green points, left plot), and once against the MB11 antibody staining, which labels GTP dimers (blue dots, right plot). Although the data points are fairly scattered, an increased Spearman correlation was found between SH and MB11 ($r = 0.56$) compared with α - tubulin (0.29). Furthermore the power fit (red line) through the MB11 data (0.345) is closer to the theoretical 0.5 power compared to the α -tubulin data (0.10).

The scatter in the data shows that in biological samples also factors other than microtubule number are at play. There are many possible reasons (such as polarity, organization and tubulin conformation) why the actual data deviate from the theoretically predicted curve. But, as mentioned by the reviewer, also signal-to-noise issues complicate actual data fitting. Because of these reasons, we chose not to emphasize the difference between the alpha tubulin and MB11 power fits in the paper itself (the fits are only included in the responses). However if the reviewer deems this necessary we can definitely also include these plots (as a supplementary or main figure) in the paper.

4) Page 9 “The cellular process with low SHG signals clearly contained less parallel organized microtubules as compared to the parallel-aligned microtubules found in the intense SH generating process”

The authors should provide an estimation of the microtubules density based on the EM images, rather than stating “clearly contained”. A simple prediction of SHG signal level from this density would also be interesting.

We have counted the number of microtubules in both regions: 7 (parallel) microtubules present in the fiber that generated intense SH and 10 (randomly oriented) microtubules for the fiber with low SH intensity. Therefore, in a simple model that would only take into account microtubule density we would expect the opposite result, in that the SH signal intensity would be higher in the fiber that contained more microtubules and thus SH emitters. The fact that this is not the case, highlights the importance of a proper parallel organization even more. We included the microtubule count in the manuscript.

Comments on the presentation
Page 4 – define “SHIM”, or use “SHG imaging” instead.

We have used “SHG imaging” throughout the paper.

Page 7 – “In dendrites, microtubules are not uniformly polarized leading to destructive interference”

-> leading to *partial* destructive interference (suggestion)

We have followed the reviewer’s suggestion and added ‘partial’ in this statement.

Page 7 “unidirectional movement of EB3-eGFP comets (Fig. 2C)” -> Fig. 2D

Figure 2 has been changed in line with the questions of reviewer 1.

In the introduction about SHG imaging of microtubules, the authors write “SHG (...) can also be used to detect the far weaker tubulin dimer as it organizes itself in the microtubule lattice (Campagnola 2001, Stoothoff 2008, Psilodimitrakopoulos 2009). -> Other studies should probably be mentioned here, e.g. Dombeck PNAS 2003, Chi-Kuang Sun J.Struct.Biol. 2004, Olivier Science 2010.

We apologize for the oversight, these references are now included in the revised manuscript.

Along the same line, page 17-18, the authors write “we found that, apart from axons, also non-neuronal cells such as fibroblasts are capable of generating SH signals from their microtubules provided that sufficient detection sensitivity is available” -> this is not a “finding”, since it was used in previous studies such as Campagnola BJ 2002, Sun JSB 2004, Olivier Science 2010.

We agree with the reviewer that “finding” was indeed not a very good choice of words. We are indeed not the first to detect SH signals from microtubules in non-neuronal cells. SH imaging of mitotic spindles has been used in several manuscripts that are now included in the revision. However in this manuscript, referring to SH from microtubules in non-neuronal cells is not related to the mitotic spindle during cell division, but to microtubules in non-neuronal cells that are not necessarily organized with a uniform polarity such as in the

spindle. Unfortunately, this was not sufficiently clear in the first version of our manuscript, therefore we have adapted the text to distinguish our results from previous work.

REVIEWERS' COMMENTS:

Reviewer #1 (Remarks to the Author):

The authors have addressed all of my concerns. The manuscript is vastly improved and the conclusions and discussion seem appropriate for the data presented. I recommend publication.

Reviewer #2 (Remarks to the Author):

Following the editorial changes and inclusion of additional data brought about by the authors, the manuscript has been clearly improved. [While this reviewer stresses that there is as yet no proof of a direct equivalence between MB11 staining and GTP tubulin content, taken together the arguments supporting an important role for GTP tubulin in SHG generation are solid].

In its present form the paper should be published and will provide important new information to scientists in the field.

In this regard, the graphs relating SHG power to alpha-tubulin and MB11 staining intensity (provided in replies to the other reviewers) are interesting even if noisy; , I suggest including them as supplementary data.

Reviewer #3 (Remarks to the Author):

The authors have addressed my previous concerns.

Point to point answer to the reviewers. NCOMMS-18-37540A

We would like to thank all three reviewers for their critical and constructive review of our paper.

Reviewer 1:

Thank you very much.

Reviewer 2:

Thank you for your comments and appreciation. As suggested, we have included the power fits as Supplementary figure 2 in the manuscript.

Reviewer 3:

Thank you very much.

Pieter Vanden Berghe